# LILA: Language-Informed Latent Actions

**Siddharth Karamcheti**[*], **Megha Srivastava**[*], **Percy Liang, Dorsa Sadigh**
Department of Computer Science, Stanford University
{skaramcheti, megha, pliang, dorsa}@cs.stanford.edu

**Abstract:** We introduce Language-Informed Latent Actions (LILA), a framework for learning natural language interfaces in the context of human-robot collaboration. LILA falls under the shared autonomy paradigm: in addition to providing discrete language inputs, humans are given a low-dimensional controller – e.g., a 2 degree-of-freedom (DoF) joystick that can move left/right and up/down – for operating the robot. LILA learns to use language to *modulate* this controller, providing users with a language-informed control space: given an instruction like "place the cereal bowl on the tray," LILA may learn a 2-DoF space where one dimension controls the distance from the robot's end-effector to the bowl, and the other dimension controls the robot's end-effector pose relative to the grasp point on the bowl. We evaluate LILA with real-world user studies, where users can provide a language instruction while operating a 7-DoF Franka Emika Panda Arm to complete a series of complex manipulation tasks. We show that LILA models are not only more sample efficient and performant than imitation learning and end-effector control baselines, but that they are also qualitatively preferred by users.[1]

**Keywords:** Language for Shared Autonomy, Language & Robotics, Learned Latent Actions, Human-Robot Interaction

## 1 Introduction

Nearly a million American adults live with physical disabilities, requiring assistance for everyday tasks like taking a bite of food, or pouring a glass of milk [1] – assistance that robots could provide. Paradigms for efficient human-robot collaboration that strike a balance between robot autonomy and human control such as *shared autonomy* [2, 3, 4, 5] present a promising path towards building such assistive systems. Unlike full autonomy approaches that enforce a sharp separation between user intent and robot execution, falling prey to problems of sample efficiency and robustness, shared autonomy couples a human's input with automated robot assistance. Consider a kitchen or dining environment where a *high-dimensional* (high-DoF) robot such as a wheelchair-mounted manipulator aids a human who may be physically unable to perform tasks requiring fine-grained manipulation. While the human can manually teleoperate the arm by fully controlling individual "modes", or separate degrees-of-freedom of the robot's end-effector, past work has shown this to be unintuitive, slow, and frustrating [3, 6]. Shared autonomy approaches such as *learned latent actions* [5, 7, 8] however, build intuitive low-dimensional controllers for high-DoF robots via dimensionality reduction.

Specifically, learned latent actions models learn *state-conditioned* auto-encoders directly from datasets of (state, action) pairs; the encoder takes the current state and action, and compresses it to a latent action $z$ with the same dimensionality as the human control interface (e.g., 2-DoF). This is fed to a decoder to try to reconstruct the original high-dimensional action. While these approaches are reliable and sample efficient, they are limited by their reliance on just the current state: given tasks like "grab the milk" and "shift the milk to the side" that overlap in state space, these controllers fail as the models lack sufficient information to disambiguate behavior.

---

[1]Additional visualizations and supplemental experiments can be found at the following webpage: https://sites.google.com/view/lila-corl21. Code can be found here: https://github.com/siddk/lila.

5th Conference on Robot Learning (CoRL 2021), London, UK.

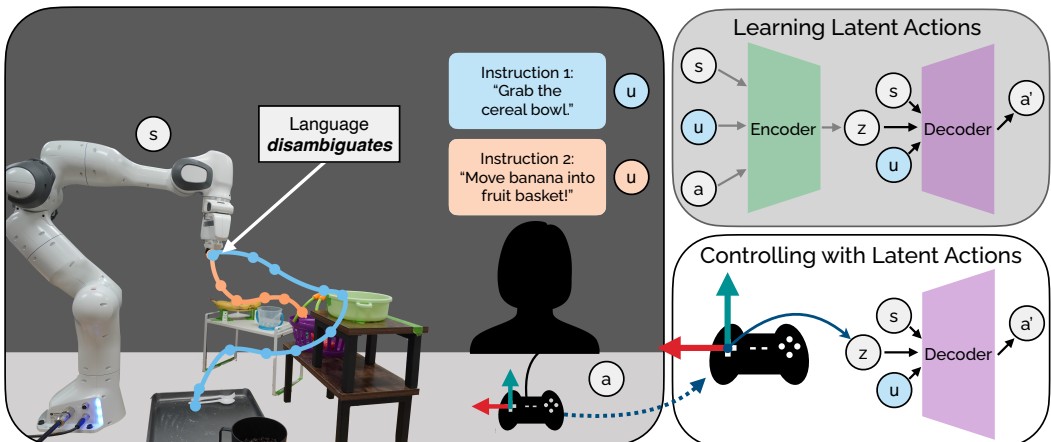

Figure 1: [**Left**] Our breakfast buffet environment with several diverse manipulation tasks. By providing both natural language input and low-dimensional joystick control [**Middle**], users *disambiguate* between different tasks while retaining the ability to maneuver through the environment. This is enabled by our [**Right**] language-informed latent actions (LILA) models that use auto-encoders to learn language & state-conditioned low-DoF latent spaces for meaningful control.

To address this concern, we consider incorporating natural language within this framework to add an additional conditioning variable for structuring the control space. Prior works integrate language in robotics settings for similar purposes within the full autonomy paradigm [9, 10, 11, 12, 13]. Unfortunately, these approaches suffer from poor sample efficiency, failure recovery, and generalization; many issues that shared autonomy methods including learned latent actions seek to address. By joining language and latent actions, a user can express an utterance $u$ = *"grab the cereal bowl"* and obtain a control space that is both state and language conditioned (Fig. 1 [Right]).

We introduce Language-Informed Latent Actions (LILA), a framework for incorporating language into learned latent actions. Key to LILA is the principle that language *modulates* a user's low-level controller based on their provided utterance; as intuition, given the utterance *"grab the cereal bowl"* as in Fig. 1, our assistive robot might learn a semantically meaningful, low-dimensional (2-DoF) control space where one dimension (one joystick axis) may control the distance from the robot's end-effector to the cereal bowl, whereas the other might control the angle of the end-effector relative to the bowl such that the robot's gripper can obtain a solid grasp of the object. Other utterances can modulate the controller in similar ways – "pour the milk into the cup" might result in a learned control space where one joystick dimension controls the jug's pouring angle, while the other may control its height. Language not only serves as a natural means for a human to communicate their intent to the robot, but also helps *disambiguate* across a wide variety of objectives as well, by inducing language and state-conditioned control spaces.

A core part of our method is its ability to handle diverse, realistic language. To this end, we collect a small, crowdsourced dataset of natural language descriptions to describe each of our training demonstrations; we use this real, natural language as the only input while training our models. To allow for out-of-the-box generalization to novel user utterances such as those that describe similar behaviors but with different words or phrases, we tap into the power of pretrained models [14, 15]. We perform a *comprehensive user study* across 10 users who use natural language and our learned LILA controllers to complete a variety of diverse manipulation tasks in a simplified assistive "breakfast buffet" setting. Our results show that LILA models are not only more reliable, performant, and sample efficient than fully autonomous imitation learning and fully human-driven end-effector control baselines, but are qualitatively preferred by users as well.

## 2   Related Work

We build LILA within a shared autonomy framework [2, 16], applied to assistive teleoperation [17, 18]. We additionally build off of work at the intersection of language and robotics [19, 20, 21].

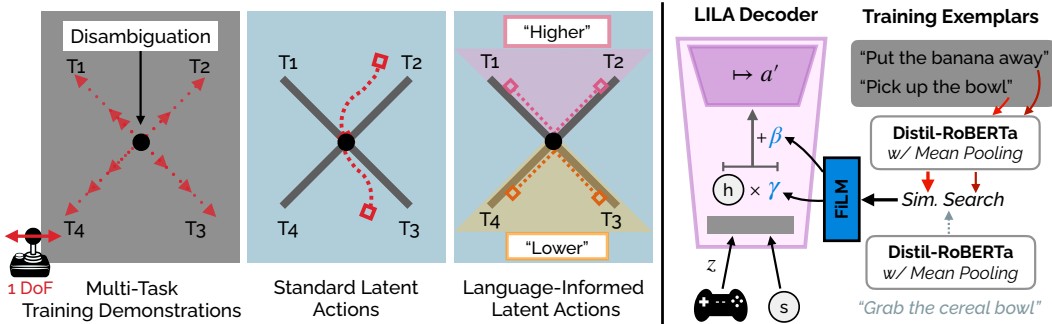

Figure 2: [**Left**] Stylized example: navigating toward four points on a cross with a 1-DoF latent action, where disambiguation is required. Standard latent action models fail, while LILA accurately reaches the corners with the help of language. [**Right**] LILA decoder architecture. We embed an utterance using a pretrained language model, then identify the closest exemplar in the training set via similarity search. We feed the embedding for this exemplar through a feature-wise linear modulation, or FiLM [46], layer that fuses language and state representations within the decoder.

**Shared Autonomy & Assistive Teleoperation.** Shared autonomy casts robot control as a collaborative process between humans and robots [2, 4, 16, 22]. While other work focuses on "blending" human inputs with possibly task-agnostic policies within the same action space [23, 24], in this work, we focus on assistive teleoperation, where a user is provided a low-dimensional controller (e.g., a joystick, sip-and-puff device) to directly control a high-dimensional robot manipulator. Using these controllers for end-effector control – e.g., via operational space control [25] – is incredibly difficult, requiring frequent mode-switching to control specific robot DoFs [3, 26]. Instead, we adopt *learned latent actions* [5, 7, 8, 27, 28] a framework that uses conditional auto-encoders [29] to learn task-specific latent "action" spaces from demonstrations. These latent spaces match the dimensionality of the low-DoF interface and provide semantically meaningful control. However, existing methods fail to differentiate between tasks with overlapping states, hindering the ability to perform diverse behaviors in a workspace (e.g., manipulating a jug of milk in different ways – pouring, placing in the fridge, etc.). In this work, we use language for *disambiguation*; users naturally speak their intent, conditioning latent action models to produce intuitive control spaces that align with user objectives.

**Language-Informed Robotics.** A variety of methods have sought to combine language and robotics, spanning approaches that map language to planning primitives [9, 30, 31, 32, 33], perform imitation learning from demonstrations and instructions [11, 34, 35, 36], and pair language instructions with reward functions for reinforcement learning [12, 37, 38]. Other approaches use language in more nuanced ways, such as learning language-conditional reward functions directly [39, 40, 41], or within adaptive frameworks, where language is used to correct or define new behavior [42, 43]. This list is not exhaustive; we present further discussion – including approaches that combine language with other modalities – in the supplementary material. However, all these approaches fall within full autonomy: after providing an instruction, human users cede control over to the robot policy, which then takes the actions necessary to perform a task.

While robots trained with these approaches can perform diverse tasks and generalize to new instructions, it is not without cost. Paramount is sample efficiency; imitation learning approaches often require hundreds to thousands of demonstrations for learning to navigate [35, 44], and reinforcement learning approaches can require millions of episodes of experience to learn robust policies [12, 45]. Whereas coarse behaviors are easy to learn, learning to recover from slight deviations from the training data, or to perform precise motions in a sequence, is incredibly difficult. By casting our approach, LILA, within a shared autonomy framework instead, we intelligently offload these parts that are harder for robots – but easier and intuitive for humans – onto the user.

## 3 Formalizing Language for Assistive Teleoperation

**Formalism.** We formulate a user's objective, or task, (on a *per-user basis*) as a fully-observable, language-augmented, Markov Decision Process (MDP) $\mathcal{M}$ defined by the tuple $(\mathcal{S}, \mathcal{U}, \mathcal{A}, T, R, \gamma)$

similar to prior work in language-conditional robotics [10, 34, 42]. Let $u \in \mathcal{U}$ denote a user's language utterance provided at the start of each episode, where $\mathcal{U}$ is the full set of language utterances a user could provide to a robot. Let $s \in \mathcal{S} \subseteq \mathbb{R}^n$ be the robot's state, and $a \in \mathcal{A} \subset \mathbb{R}^m$ be the robot's action: taking action $a$ in state $s$ results in a next state $s'$ according to the transition function $T(s, a)$. Given the language utterance $u$, the user implicitly defines a reward function $R(s, u, a) \in \mathbb{R}$; the human and robot collaboratively maximize this reward subject to discount factor $\gamma \in [0, 1]$.

**Problem Statement.** This MDP forms the basis of a shared autonomy task wherein a human is equipped with a low-dimensional control interface for the robot. Let $z \in \mathcal{Z} \subset \mathbb{R}^d$ where $d \ll m$ be the human's control input to the robot, such as the $d = 1$ DoF controller in Fig. 2. Previous work on learning latent actions for assistive teleoperation [5, 28] learn a decoder $\mathrm{Dec}(s, z) : \mathcal{S} \times \mathcal{Z} \rightarrow \mathcal{A}$ that maps user low-dimensional inputs $z \in \mathcal{Z}$ and current state $s \in \mathcal{S}$ to a high-dimensional action $a \in \mathcal{A}$. However, in situations where state-conditioning is not enough to disambiguate a users' intent, too low of a control input dimension $d$ may lead to failure. Recalling the milk jug example, we have multiple different behaviors we could execute if the end-effector were next to the milk. For example, one might want to pick up the jug, shift it to the side, pour it, etc; conditioning only on the state with a 2-DoF action space is not enough to recover all possible behaviors.

Instead, we aim to learn $\mathrm{Dec}(s, u, z) : \mathcal{S} \times \mathcal{U} \times \mathcal{Z} \rightarrow \mathcal{A}$ that takes the user's control input *and* utterance $u$, and predicts the high-DoF action that matches the user's objective. The utterance $u$ acts as additional conditioning information, producing control spaces that depend on both language and state; this circumvents the disambiguation problem above.

## 4 Language-Informed Latent Actions (LILA)

We are given a dataset of demonstrations, where each demonstration contains an utterance $u$ and a trajectory $\tau = \{s_0, a_0, s_1, a_1 \ldots s_T\}$. We split each demonstration into triples of $(u, s_i, a_i)$ and use these to learn a conditional auto-encoder, consisting of a language-conditional encoder Enc: $\mathcal{S} \times \mathcal{U} \times \mathcal{A} \rightarrow \mathcal{Z}$ that maps to a latent $z$ and a decoder Dec: $\mathcal{S} \times \mathcal{U} \times \mathcal{Z} \rightarrow \mathcal{A}$ that attempts to reconstruct the original action $a$. We minimize the mean-squared error between the predicted and the original action:

$$L_{\mathrm{Enc,Dec}} = \frac{1}{N} \sum_{i=1}^{N} (\mathrm{Dec}(s_i, u_i, \mathrm{Enc}(s_i, u_i, a_i)) - a_i)^2 \tag{1}$$

We next discuss how we integrate language into the architecture of the encoder and decoder.

### 4.1 Integrating Language within the Latent Actions Architecture

We implement the encoder and decoder as multi-layer feed-forward networks, with the ReLU activation as in prior work [7, 28]. We focus on the decoder here, but the encoder is symmetric. For the decoder, we first concatenate the robot state $s$ and latent action $z$, then feed the corresponding vector through multiple ReLU layers (usually 2-3), upsampling to produce the high-dimensional robot action. We next discuss how to incorporate language within this simple scaffold.

Pretrained language models such as BERT, T5, and GPT-3 [47, 48, 49] have revolutionized NLP, providing powerful language representations. Inspired by their success when applied to robotics and reinforcement learning tasks [38, 50, 51], we use a distilled RoBERTa-Base model [14], from Sentence-Transformers [15] to encode utterances. This model is fine-tuned on a corpus of paraphrases, allowing it to pick up on sentence-level semantics. We generate utterance embeddings by performing mean-pooling over token embeddings for an utterance, as in prior work [38, 50]. We incorporate these embeddings using feature-wise linear modulation (FiLM) layers [46] that fuse language information with other features $h$ by mapping language embeddings to parameters $(\gamma, \beta)$ of an affine transformation: $h' = \gamma * h + \beta$ (Fig. 2). Notably, this $h$ is the representation received after feeding the state and latent action $(s, z)$ through the *first* layer of the decoder as described above. Once the language transforms $h \rightarrow h'$, we feed $h'$ to the subsequent layers of the decoder.[2]

---

[2]Implementation can be found in the open-source code repository: https://github.com/siddk/lila.

| Task Name | Success | Example User Study Input | Mapped Training Data |
|---|---|---|---|
| `Pick Banana` | 100% | *yellow in purple* | → `pick up the yellow banana and place it into the purple basket` |
| `Pick Fruit Basket` | 100% | *bring basket to center of pan* | → `place the basket onto the tray` |
| `Pick Cereal` | 100% | *go to the left side of the cream bowl, go down, grab the cereal bowl, and place it on the try* | → `grab the cereal bowl and put it on the tray` |
| `Pour Bowl` | 67% | *pick up the cup of marbles and pour them into the cereal bowl* | → `pick and pour the cup of white balls into the bowl of cereal` |
| `Pour Cup` | 100% | *pick up the clear cup with marbles in it and pour it in the black mug with the coffee beans in it* | → `pick up the cup and pour the contents in the mug` |

Table 1: Example utterances provided by study participants paired with the retrieved exemplar per §4.1. Success rate refers to the percentage of the time a user study utterance (over all utterances in the study) was grounded to the correct task via our retrieval method. As each participant only attempted 2 tasks, success rate can fluctuate significantly, as is the case with the `Pour Bowl` task.

**Nearest-Neighbor Retrieval at Inference.** A major concern for work in language-conditioned robotics is generalizing to novel language inputs. While it may be unreasonable to expect generalization to completely new tasks, for user-facing systems with a clear set of behaviors seen at train time as in our work, there is an expectation that any language-informed system is capable of handling moderate variations of utterances from the training set. To do this, adding linguistically diverse data has been the gold standard [44, 51]; however, a new class of approaches have emerged that sidestep additional data requirements by tapping into the potential of pretrained language models [43, 50]. These approaches frame language interpretation at inference, when interfacing with real users, as a *retrieval* problem: each new user utterance $u'$ is embedded(with the same pretrained model as above, then used to query a nearest neighbors store containing all training exemplars; once the nearest neighbor $u_i$ has been identified, it replaces $u'$ as an input to LILA.

The key benefit of such an approach is the minimal mismatch between train and test language inputs: all "test inputs" are drawn from the training set. This does mean, however, that user utterances that describe new tasks, or are otherwise unachievable also get mapped to language seen at training. While this limits the ability to perform novel tasks, it again highlights the benefits of the shared autonomy paradigm – doubly so, considering the cost of a mistake in an assistive domain like the one we consider in this work: if, while providing control inputs to the robot, a user feels the robot is not acting in alignment with the user's desired objective, they can always stop execution.

## 5  User Study

We evaluate LILA with a real-world user study on a 7-DoF Franka Emika Panda Arm, on a series of 5 complex manipulation tasks. Each user is provided a 2-DoF joystick for control. We compare against a non-learning, end-effector (EE) control baseline where users "mode switch," controlling the velocity of 2 axes of the end-effector pose at a given time – [(X, Y), (Z, Roll), (Pitch, Yaw)]. Language utterances $u$ are typed into a text console for simplicity; future work could extend this work by using off-the-shelf speech recognition systems. We also compare against a fully autonomous imitation learning (IL) strategy where users solely provide language inputs, and the robot attempts to perform a task without additional input. Ostensibly missing is a *no-language* variant of the latent actions model, in keeping with prior work; however, upon evaluating this model, we found it to be unintuitive and unable to make progress or solve any task, so we omit it from our user study. However, further experiments and analysis can be found in the supplementary material.[3]

---

[3]Experiments with the no-language baseline and extra analysis showing the necessity of extra conditioning information can be found here: https://sites.google.com/view/lila-corl21/home/no-lang-baseline

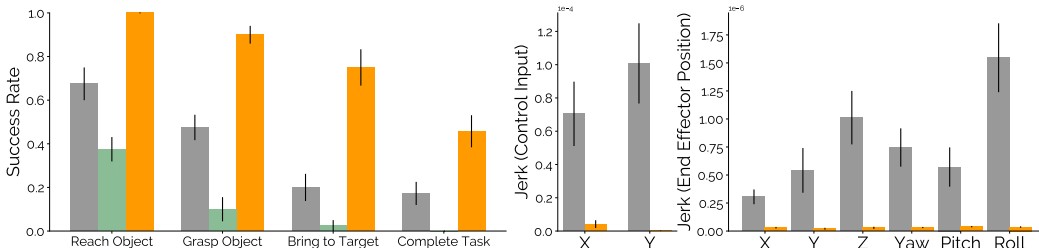

Figure 3: Quantitative Results. We average success rate [**Left**] across all sub-tasks for each control method, and find that LILA is significantly ($p < 0.05$) more performant. However, the steep drop in performance when completing the full task shows the difficulty of fine-grained control. We also calculate jerk as an indicator of controller smoothness, for both user control inputs [**Middle**] and end-effector position [**Right**]. Averaged across tasks and users, we find LILA leads to significantly smoother control for users than end-effector control.

**Environment.** Fig. 1 shows our "breakfast buffet" setting, a scaled down version of an assistive feeding domain. We define 5 tasks: 1) `Place Banana`: placing the banana in the purple fruit basket, 2) `Place Basket`: grasping the purple basket by the handles and dropping it on the tray, 3) `Place Bowl`: grasping the green cereal bowl by its edge and moving it to the tray, 4) `Pour Bowl`: pouring the blue cup of marbles (a proxy for milk) into the cereal bowl positioned on the tray, and 5) `Pour Cup`: pouring the blue cup of marbles into the yellow coffee cup. Fig. 1 shows idealized example trajectories for the `Place Bowl` (blue) and `Place Banana` (orange) tasks.

These tasks vary in difficulty, requiring precise grasping and dexterous manipulation. We evaluate partial success based on how many of the following 4 subtasks users are able to complete: 1) `Reaching`: touching the desired object, 2) `Grasping`: executing a successful grasp, 3) `Bring to Target`: successfully transporting the manipulated object, and 4) `Task Completion`.

**Demonstration Collection.** Both LILA and IL models require learning from (language, demonstration) pairs for all 5 tasks. We collect demonstrations kinesthetically as in prior work [7, 28], recording joint states at a fixed frequency. We initially collected 15 demonstrations per task for each method. However, on testing the IL model, we found it incapable of performing even rudimentary reaching behaviors. To give IL the best chance, we collected twice the number of demonstrations (30 per task; 150 total), requiring an extra 2 hours of labor.

**Crowdsourcing Language Annotation.** To build a *natural* language interface for human-robot collaboration, we collect language annotations for each task by crowdsourcing utterances. Our goal was to capture the diverse ways users may refer to the objects and actions our tasks entail without any additional information, simulating a real user interacting with our environment for the first time. We recruited 30 workers on Amazon Mechanical Turk, showing only a video of a recorded demonstration, and asked them to provide *"a short instruction that you would want to provide the robot to complete this task independently in the box below."*. However, this procedure resulted in some annotations containing "spam", or extremely out-of-domain text. To address this without introducing our own bias on what constitutes "spam", we filtered the data to identify workers who consistently provided "noisy" annotations, measured by the cosine distance between the sentence embedding (using any pretrained embeddings) of an annotator's provided text and the *average* sentence embedding aggregated over all other annotators for a given video. We used annotations from the 15 least "noisy" annotators under this metric as our ground-truth utterances. Further details, as well as example "spam" annotations that were filtered out, are in the supplementary material. Table 1 provides examples of crowdworker utterances from our final dataset (rightmost column).

**Participants & Procedure.** We conducted our study with a participant pool of 10 university students (5 female/5 male, age range $23.2 \pm 1.87$). Four subjects had prior experience teleoperation a robot arm.[4] We conduct a within-subjects study, where each participant completed 2 tasks, chosen

---

[4]Due to the COVID-19 pandemic and university restrictions, only those with pre-authorized access could participate. See the COVID-19 considerations document in the supplementary material for more details.

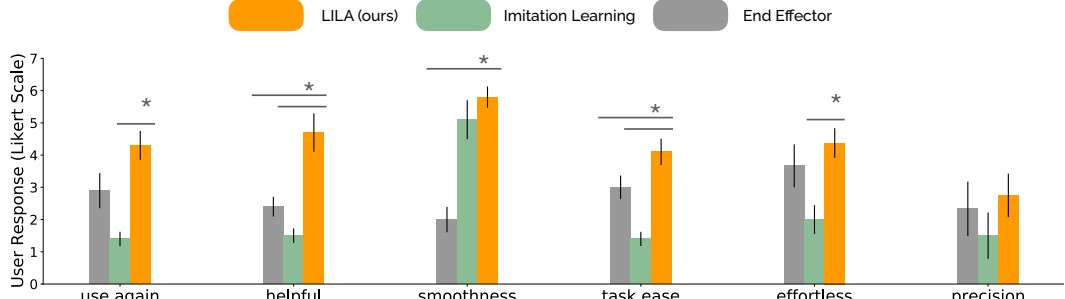

Figure 4: Qualitative Results. Using a 7-point Likert scale, we ask users to evaluate each of the 3 control methods for different properties. With high significance ($p < 0.05$), we find that LILA outperforms both imitation learning and end-effector control baselines on several metrics, including degree of helpfulness provided and ease in completing tasks.

randomly, with each of the three methods. Users were given 2 trials to complete each tasks, and an allotted 3 minutes per control strategy to practice. Users were also given a sheet describing controller inputs and details for each control method, which we include in the supplementary material. For imitation learning and LILA controllers, which require language inputs, participants provided a natural language utterance which which a proctor entered into the model – participants were allowed to verify the proctor entered their input accurately. This user-provided language utterance is used as the query in the nearest-neighbor retrieval described in §4.1; the retrieval set consists of all training utterances collected via the crowdsourcing procedure above. In addition to tracking quantitative success rates (normalized, based on progress relative to each of the 4 defined subtasks), time taken per task, and controller logs, we ask users to fill out a qualitative survey evaluating each method at the end of each study. We present both quantitative and qualitative results below.

**Quantitative Results.** Fig. 3 summarizes our objective results. We evaluate both full- and partial-task success rates for each task across all control methods, in addition to computing smoothness metrics directly on the logged user inputs and robot actions. Smoothness is a measure for intuitiveness when measured on user 2-DoF joystick inputs, ease of use when measured on the robot's end-effector pose, and implicit safety: a trajectory with high discontinuity in acceleration can lead to rapid, unpredictable changes in the environment. Smoothness is negatively correlated with *jerk*, the time-derivative of acceleration. We compute jerk by taking the second-order derivative of velocity, and report average jerk across fixed windows. Our results show that LILA significantly ($p < 0.05$) outperforms both methods across all sub-tasks, and is also smoother to use both in control input space (2-DoF input) and end-effector space (6-DoF). However, the relative drop in performance of LILA for the final sub-task "Complete Task" shows the room for improvement in fine-grained control, such as pouring motions. Additionally of note is the poor performance of imitation learning. To explore this fully, we perform an ablation, and show that sample inefficiency is a likely cause – especially since we are in a low-data regime. These results and arguments can be found in the supplementary material.[5]

**Qualitative Results.** Fig. 4 summarizes our subjective results. We administered a 7-point Likert scale survey after users finished performing tasks with each method; this survey included questions around the perceived helpfulness of the model in completing the tasks (*helpful*) and whether the participant would use the control method again (*use again*). The results show that LILA outperforms both imitation learning and end-effector control across most qualitative metrics, with significant results ($p < 0.05$) marked with an ∗. We additionally visualize samples of the observed end-effector trajectories by individual users collected during our study for 3 of our 5 tasks in Fig. 5. Across all tasks, LILA results in smoother end-effector trajectories than end-effector control, while imitation learning comes close to the target object but is unable to complete the entire trajectory for the task.

---

[5]Additional experiment videos, and a results of the imitation learning ablation can be found here: https://sites.google.com/view/lila-corl21/home/il-ablation

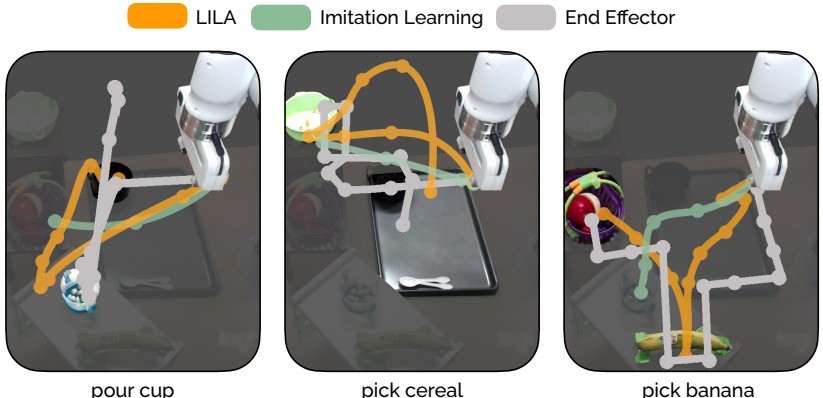

Figure 5: Trajectories from one user comparing three different control methods 3 out of our 5 tasks – **Pour Cup**, **Pick Cereal**, and **Pick Banana**. LILA provides smooth actions that immediately approach the target object for a task, while end-effector control shows more rigid motions that result in users diverging from their intended paths. While imitation learning also enables smooth motions, it often fails shortly after reaching objects, hence the shortened trajectories.

## 6   Discussion

**Summary.** We present Language-Informed Latent Actions (LILA) a framework that marks the first step in combining the expressiveness and naturalness of language for *specifying* and *executing* on a human user's objective within the context of assistive teleoperation. Our user study results show that when compared to fully autonomous, imitation learning approaches, LILA is more sample-efficient and performant, training on half the number of task demonstrations, but obtaining significantly higher success rates. Compared with no-learning end-effector control methods, we again show LILA's effectiveness at obtaining high success rates, but also demonstrate its ability to produce intuitive low-dimensional control spaces from language input. Qualitatively, we find that users prefer LILA to alternative methods across the board, opening the door for additional work on language & latent actions.

**Limitations and Future Work.** Currently, LILA uses language as a mechanism for *task disambiguation* – in the current results, there is no mechanism for generalizing to completely unseen tasks or language specifications. We believe that the ability to disambiguate with language, and the integration of language within the latent actions framework is a strong research contribution, and hope that future work looks to dynamic states – perhaps by leveraging visual latent actions [28] – and to adapting to new utterances and tasks dynamically [43, 52]. Furthermore, while users found LILA intuitive and natural, they found themselves wanting to further modulate the robot's behavior with language instructions *during the course of execution*. Many users, upon seeing the robot make slight deviations from a desired path would instinctively provide spoken corrections – *"a little to the right"*, *"no, grasp it by the handle!"* – indicating a desire for *multi-resolution* language control.

**Shared Autonomy and LILA.** LILA fits within the shared autonomy paradigm, where the role of language has been underexplored. With LILA and shared autonomy approaches in general, humans retain *agency* – they are responsible for robot motion, and if the robot moves in a way that is not safe, or does not align with their objectives, they stop providing control and possibly reset, give a new instruction, or drop into a more complex control mode. Behavior is *interpretable* – the latent actions model, *critically informed by language*, produces intuitive control spaces that humans can quickly grasp. Finally, language is *natural* – users specify their objectives as they would if speaking to another person, and the robot uses that language to shape their control space. These properties – preserving agency, maintaining interpretability, and leveraging the expressive and natural features of language for specifying objectives – are critical for widespread human-robot collaboration, and we hope this work presents a concrete step towards achieving that goal.

**Acknowledgments**

Toyota Research Institute ("TRI") provided funds to assist the authors with their research but this article solely reflects the opinions and conclusions of its authors and not TRI or any other Toyota entity. This project was also supported by NSF Awards 2006388 and 2132847. Siddharth Karamcheti is grateful to be supported by the Open Philanthropy Project AI Fellowship. Megha Srivastava is supported by the NSF Graduate Research Fellowship Program under Grant No. DGE-1656518.

We would additionally like to thank the participants of our user study. We further extend a special thank you to Madeline Liao and Raj Palleti for helping with the visual intuition for our imitation learning analysis. Finally, we are grateful to our anonymous reviewers, Suneel Belkhale, Ajay Mandelkar, Kaylee Burns, and Ranjay Krishna for their feedback on earlier versions of this work.

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
