# OpenReview forum: "LILA: Language-Informed Latent Actions"
_robot-learning.org/CoRL/2021/Conference — CoRL2021 Poster_

### Official Review · Reviewer_Eg7K · 2021-07-06

**Originality:** Very Good
**Technical Quality:** Excellent
**Clarity Of Presentation:** Very Good
**Impact:** 4

**Recommendation:**

Strong Accept: I recommend accepting the paper and will argue for my recommendation even if other reviewers hold a different opinion.

**Summary:**

This paper presents a shared autonomy framework that reduces full robot arm control to 2 degrees of freedom, deciding along what control axes to align those degrees based on a natural language input command for the task to be completed, such as "pour the milk into the bowl". The method presented uses a transformer-based, massively pretrained language model to perform nearest neighbor lookup at inference time to map language commands to understood data points from the training set to modulate control. The work is a crisp step towards a larger space of language-guided shared autonomy.

*Post-response*

The authors substantially revised the paper in response to other reviewer's concerns. The main concern I had regarding data filtering was clarified as an MTurk cleanup procedure. My only comment for future work from the authors there is that a second "filtering" HIT is a cleaner way to do this. Basically, gather data D_0, run a filtering HIT where annotators select whether instances in D_0 are valid (e.g., are they spam, do they match a demonstration, etc.), then throw out bad D_0 instances *and* remove worker qualifications for the annotators who provided them, then repeat the data collection step to fill out D_1 replacing now-missing instances. Repeat to D_i full dataset.

**Issues:**

Questions by priority

Q1 In "Crowdsourcing Language Annotation", annotations whose representations fall away from a mean are discarded, which feels a bit icky because the method later uses those representations to do nearest neighbor lookup. It could be argued, then, that the data has been curated for the algorithm, keeping around only a local neighborhood of utterances to retrieve without allowing for outliers that could cause trouble at inference time. Can the authors clarify this methodology?

Q2 For Figure 4, I was surprised to see End Effector performing better than Imitation Learning in some cases. Is there a clear intuition for this result? Maybe IL just needs a lot more demonstrations before the policy isn't too frustrating?

Nits and questions not effecting evaluation:

- Line 77 "unprompted" unclear. Does this mean some users didn't use language at all?

- Line 166 "action" in quotes is a weird way to describe space Z since there is already an action space; the paper knows this since it's in scare quotes, but maybe just use a different word.

- Line 192 Grounding Language in Play also pulls this "retrieve utterances from training time" trick, if I recall correctly.

- Line 250; 5 students were female, were the other 5 male? Leaving this inference implicit erases other gender identities and saves almost no space in the paper.

- Line 289 "heralding a bright future" https://i.gifer.com/Dnxs.gif

**Reviewer Expertise:**

Good: General knowledge of the area

**Strengths And Weaknesses:**

The main strength of the paper is a clear method for langauge-informed shared autonomy with convincing human subject experiments for a physical robot agent.

The main weakness of the paper is not having a head-on discussion about exactly what kind of generalization from training data is possible under LILA's current formulation. In particular, the model can "sort of" generalize to unseen language commands, as long as they are paraphrases of commands seen at training time. To my understanding, the model has no capacity to generalize to unseen tasks and entirely unseen language specifications for those new tasks.

**Summary Of Recommendation:**

The paper makes a concrete step in RoboNLP towards shared autonomy and contains convincing, real-world experiments with humans controlling a robot arm conditioned on free form language input. When I imagine the intersection of "RoboNLP" and "CoRL", this paper is a great example.

---

> ### Author Response · Authors · 2021-08-31
> **General Response [Eg7K]**
>
> Thank you for your thorough review, and for believing in our work. We hope that our above comments and updated paper solidify your position on our paper. Please let us know if there is anything else we should include for the final version!
>
> To respond to your open questions:
>
> > *“The main weakness of the paper is not having a head-on discussion about exactly what kind of generalization is possible under LILA’s current formulation”*
>
> This is completely fair. You’re correct in that LILA has no current mechanism for generalizing to unseen tasks and completely unseen language specifications. We have made this more explicit in Section 6 — “Limitations.” To provide a slight justification (though we believe you know this already) — incorporating language for just disambiguation in the latent actions paradigm was an unknown from a research perspective; we weren’t sure if we could use language to learn multiple task-specific control spaces, how much data it would require, and so on. Now that we’ve shown the possibilities just with disambiguation, we hope that future work will go the way of different types of linguistic and environment generalization, including handling shifting environments, different types of linguistic generalization at multiple resolutions (“get the cup...”, “no the one on the left”) as well as compositional and few shot generalization.
>
> Generally, we’re very excited at how sample efficient and real-robot ready this approach is, and hope this sort of problem formulation becomes popular in RoboNLP more generally — as we think many of the questions that are posed in the field can be explored here practically, on real hardware.
>
> > *“Q1. Crowdsourcing Language Annotation... can the authors clarify this methodology?”*
>
> We hope that the clarification top-level comment above addresses this point. We’re really just filtering to remove SPAM, and not really for curation; if you’d like additional experiments to this effect in the final paper please let us know.
>
> > *“Q2. I was surprised to see End-Effector performing better than Imitation Learning in some cases... maybe IL just needs a lot more demonstrations?”*
>
> We hope that the top-level comment on the Imitation Learning baseline helps clarify this. Basically, you’re totally right about the sample-efficiency of Imitation Learning, and this is further explained in the technical argument on the attached webpage. We hope that this is clear, and especially given the extremely low-data regime (15 demos per task!) we’re in, this makes sense! Let us know if there’s anything further you’d like to see in the final paper.
>
> > *Nits*
>
> We have addressed these in the main paper. “unprompted”just means that we did not coach our users to use specific words or instructions — we see this is confusing so we removed this. We've refered to the “action” space (Z) as the latents or “latent action space” to avoid confusion. We did not mean to erase other gender identities; we apologize and have fixed this in the paper. “Heralding” —> we’ve toned it down :).

---

### Official Review · Reviewer_o584 · 2021-07-23

**Originality:** Good
**Technical Quality:** Good
**Clarity Of Presentation:** Very Good
**Impact:** 3

**Recommendation:**

Weak Accept: I recommend accepting the paper, but will not argue for my recommendation if the majority of other reviewers have a different opinion.

**Summary:**

This paper presents an approach to improving shared autonomy in which a user controls a high degree of freedom robot arm with a low dimensional controller and provides natural language instructions to disambiguate user intention. Building on recent work in learning latent actions for shared autonomy---for which language has not previously been used---the authors demonstrate the utility of adding language on a series of manipulation tasks. The experiments in the paper demonstrate that their approach, language-informed latent actions or LILA, allows the human-robot team to achieve good performance on the selected tasks, outperforming two baseline strategies.

**Issues:**

Much of the rationale for my decision and my suggestions for changes are included in the "Strengths and Weaknesses" section above, and so the authors should refer to that for more detail. In short, the lack of a baseline comparison algorithm that uses non-language-informed learned latent actions is conspicuous; the authors should provide justification for why this baseline was not included and why the central claims of the paper are sufficiently supported by the experiments that currently exist.

EDIT (post rebuttal): See my comments above, but I believe that the addition of the no-language baseline satisfied my primary concern. I have updated my score to reflect this change.

**Reviewer Expertise:**

Good: General knowledge of the area

**Strengths And Weaknesses:**

Overall, the method is innovative and the paper is well written. However, it is the central claim of the paper is that the addition of language is necessary to disambiguate the robot's actions when trying to tackle a task in the presented shared autonomy scenario, yet the results (though otherwise informative and convincing) lack a comparison to a learned method that does not use language. Such a comparison is important to back up their central claim and show that the language input is indeed necessary for this situation. This omission is particularly surprising, as the primary point of comparison made in the remainder of the paper (e.g., the theory section) is to previous work on non-language-informed latent action learning. While the results are shown to work and the users are able to accomplish their tasks, it is not clear that it is the addition of the language that make this feat possible, greatly weakening the paper. The comparison to the Imitation Learning baseline, while informative, does not allow for a similarly realistic comparison. This is the central limitation of the current version of this work, and the authors should clarify why non-language latent action experiments were not included.

EDIT (post rebuttal): While I will admit that it was a bit of a strange decision to omit the no-language baseline from the user study, I am satisfied by the experiments now included in the paper. I think this is a reasonably solid paper, made whole by the addition of the new no-language baseline. I am changing my score to reflect this.

*Minor Comments and Suggestions*
- Figs. 1 & 2: After reading through the paper, these figures are helpful visualization of the proposed approach, but on their own (without the body text), the figures do not do a particularly effective job at illustrating how everything works. Figure 1 in particular is not particularly informative if one is not already familiar with learned latent actions; some additional visualization of the user input (to show that it was shared between the blue and orange instructions) would be incredibly helpful. For Figure 2, it was not clear from the schematic that the agent could . Furthermore, the figure is somewhat misleading, since using the 1DoF controller does not necessarily correspond to moving along the left-right direction in the figure. Both figures should be revisited for clarity.
- What does "success" refer to in Table 1 (and how many trials are included in each; I could not find this information in the text)? Is it the success of the retrieval or of the task associated with the specified task. Relatedly, what does a failure "look like" for the Pour Bowl task?
- There is a typo in Figure 3: The success rate is presented as a percentage, yet the y-axis only goes to 1.0%. Presumably, this is 100% success.
- A minor point: while not a style guide violation, the abstract is quite long, and it could be improved if the authors were to shorten it somewhat.
- I believe that, owing to the "retrieval" nature of the language model, all of the tasks appear in the training dataset. It would be helpful for clarity for the authors to reiterate this point in the experiments section. [I do not think additional experiments are required.]
- Under participants, it is mentioned that 5 are female: are we to presume that the other 5 are men (or non-binary or did not specify)? It's a small change, but promotes inclusion.

**Summary Of Recommendation:**

Overall, upon first reading I was quite enthusiastic about this paper (and generally remain so) because, though the technical advance is not overwhelming, it represents solid progress in a difficult area and is supported by some good experiments that show the efficacy of using LILA, the proposed approach. However, the omission of a comparison to prior work in non-language-informed latent actions is hard to overlook. As I mention above, the central claim of the paper relies on language being a key distinguishing feature versus previous work in latent action learning, yet the only other learned baseline uses imitation learning (without subsequent human intervention) that also relies on learning. I am willing to change my recommendation if the authors can provide sufficient justification for its omission.

EDIT (post rebuttal): See my comments above, but I believe that the addition of the no-language baseline satisfied my primary concern. I have updated my score to reflect this change.

---

> ### Author Response · Authors · 2021-08-31
> **General Response [o584]**
>
> Thank you kindly for taking the time to thoroughly understand our paper. We are excited by your enthusiasm around our paper, and we hope that the above comment regarding the “lack of a comparison to a method that does not use language” (No-Language Baseline) covers many of your open questions and comments. If this is not sufficient justification (we really hope it is, and enables you to resolve your doubts and place your faith in our paper!) we’re happy to include further experiments in the final version.
>
> Now, to tackle the concerns in your review:
>
> > *“However, the omission of a comparison to prior work in non-language-informed latent actions is hard to overlook”*
>
> We hope the above comment regarding the “No-Language” baseline addresses this. To be clear, we had two figures in our supplemental that show the random, non-intuitive behavior we get from the no-language latent actions model; however, we know now that those figures themselves were not enough. We hope the above comment, paired with the [extra videos showing the non-intuitiveness of the baseline](https://sites.google.com/view/lila-corl21/home/rebuttal-no-language-baseline?authuser=0#h.rd53jubxf8w7) (and its *inability to make progress and solve any tasks*) coupled with the formal argument around the number of degrees-of-freedom required help justify the omission.
>
> Punchline: the no-language baseline trained on all demonstrations with no conditioning learned to “average” the demonstrations for the multiple tasks, thus learned a “clobbered”/”overloaded” latent space that prevented users from making *any* progress. It was a vacuous baseline, and we’re happy to show more examples in the final paper if desired. Including it in our user study felt like it wouldn’t be informative, and would add lots of time to an already long in-person study.
>
> For the minor comments:
>
> > *“Success in Table 1... what does a failure look like”*
>
> Table 1 is a bit incomplete, we add more detail in the revised text. This table encompasses the entire user study, and the language instructions provided. Success refers to how often the user-provided utterance was mapped to an annotation of the “desired” target task. A failure then, means that the user-utterance was mapped to another task (e.g., saying something with the intent of placing the banana in the fruit basket, but instead getting mapped to the control space for “picking up the fruit basket and placing it on the tray”).
>
> > *“Retrieval nature of the language model”*
>
> We have reiterated this in the experiments section — given a novel utterance from a user during the user study, we retrieve using the set of training instructions we collected for our set of training tasks.
>
> > *Other style & comments*
>
> We’ve fixed the typo in Figure 3. We’ve shortened the abstract. For participants, we did not mean to erase other gender identities; we apologize, and have fixed this in the paper.

---

### Official Review · Reviewer_TkuX · 2021-07-23

**Originality:** Very Good
**Technical Quality:** Fair
**Clarity Of Presentation:** Excellent
**Impact:** 4

**Recommendation:**

Weak Accept: I recommend accepting the paper, but will not argue for my recommendation if the majority of other reviewers have a different opinion.

**Summary:**

The paper proposes a method to learn a mapping from low-DOF (e.g. 2-DOF joystick) control inputs to joint space (e.g. 7-DOF) commands that are otherwise inaccessible or unintuitive to human operators, conditioned on a natural language task description. Given a natural language utterance describing a task (e.g. pour milk into the bowl), the system produces a control mapping that allows a human user to more easily, smoothly and efficiently complete the stated task. This is a shared autonomy as opposed to a full autonomy setting as explored in prior instruction-following works, where ultimately the control over robot motion is with the end-user.

At test-time the system uses a BERT-based sentence similarity metric to map a natural language utterance to the closest synthetic task description in the training set (seemingly among a set of 5 tasks). Then, a control decoder is conditioned on the task description and maps 2-DOF joystick commands to 7-DOF joint space.

The paper performs a user study to compare the effectiveness and user-friendliness of their approach against a fully articulated mode-switching control scheme, and also against an imitation learning instruction-following approach. The paper shows that LILA is more effective at enabling the user to accomplish manipulation tasks, achieves smoother motion, is more likely to have the users prefer to use it again, and other positive qualities.


**Issues:**

- Lines 242-246: It is not stated whether the best annotators are typical or atypical. I could imagine each having pros and cons. Typical might give more sensible and natural language. Atypical might provide more diverse training data useful to expand the scope of natural language that the system can understand.
- I would suggest trying mean-pooled GloVe vector representations to capture sentence similarity. They seem to be potentially better on simple sentences than BERT, and have the additional advantage of being lightweight (See https://arxiv.org/pdf/2011.05864.pdf).
- I suggest that slightly more model details (e.g. high-level architecture, how the features h are fed into the networks, how FiLM is used) are provided in the main paper, perhaps at the expense of the introduction, which could easily make the same points more concisely..

Suggestion for incorporating vision:
- The pre-trained CLIP feature extractor (https://openai.com/blog/clip/) is known to be useful for grounding object references to locations in the environment. Perhaps there's a way to utilize it without requiring expensive language data collection?

Minor comments:
- It’s a bit hard to understand what this sentence means: “injects linguistic information into the core of a standard latent action decoder.”
- Unless I am missing something, the reward function R(s, a, u) does not appear to be ever used as there is no reward optimization taking place. I would suggest cleaning up the formalism section so that it better aligns with the actual system under consideration.

Questions:
- (Line 150): what guarantees that the controller exposes an intuitive action space that listens to the user? I could imagine that the controller may ignore z and instead focus on inputs s,u to directly optimize R(s,a,u), since z is not strictly necessary for this purpose.
- How does the loss in Equation 1 ensure that the learned latent space is smooth and intuitive? I could imagine it having jarring discontinuities.

**Reviewer Expertise:**

Very good: Comprehensive knowledge of the area

**Strengths And Weaknesses:**

Strengths:
- Explores a novel, relevant, and interesting idea. Opens doors to exciting follow-up work.
- The approach of mapping a natural language utterance at test-time to the closest synthetic language utterance from the training set has been explored before in a full autonomy setting, where it significantly restricts the scalability of the system capabilities. This paper applies this method in the shared autonomy setting, where the user is ultimately in charge of the robot actions. Here, it makes a lot more sense. The scalability concern (e.g. of handling novel objects or composing primitives in novel combinations) is largely addressed by the human pilot, while the restricted synthetic syntax reduces the chance of an out-of-distribution utterance resulting in a noisy or unusable control mapping.
- Performs a real-world end-user study, which is both rare and necessary to assess performance of systems with natural language interfaces.
- Opens doors to interesting uses of shared autonomy besides end-user use, for example to improve the ease of crowdsourcing demonstrations on a dexterous robot arm corresponding to specific instructions.

Weaknesses:
- Unless I am missing something, the system does not take any visual inputs or observations of the environment as input. It is therefore incapable of actually understanding or grounding the natural language utterances to percepts or entities in the world. The experiments work solely because the training tasks and test tasks use the exact same objects in the same placements with no within-task variation. Therefore solving the task boils down to classifying the input utterance into one of the 5 task types, and then applying a fixed control mapping corresponding to the task type. This does not demonstrate any meaningful language grounding capabilities, which should include aspects like grounding noun phrases to visual percepts, responding to object placement variations, and allowing compositional specifications of object attributes. Any practical shared autonomy system would need to see the world and respond to it, dynamically interpreting the meaning of the instruction in relation to the objects in the environment. Prior instruction-following methods already demonstrate these capabilities, so I don’t see a strong reason to take such a significant step back in terms of the experimental setup.

**Summary Of Recommendation:**

The paper proposes a very exciting and relevant use-case of grounding natural language instructions to control mappings for a shared autonomy setting, and a user study of an initial prototype system showing effectiveness and user-friendliness in a shared-autonomy setting. This not only opens a new, exciting area of research, but is also relevant to assistive robotics, where it could significantly accelerate the adoption of robots that help people with disabilities, and bring value in the near term while fully-autonomous systems are still not robust enough.

However, as described in the “Strengths and Weaknesses”, the system does not ground language in observations of the environment, instead applying a fixed joint-space mapping that relies on objects being placed in previously known positions. This is a significant technical weakness. First, there is no meaningful language grounding or language understanding going on, and second, there does not appear to be a way to scale the system to a practical scenario, even a simple table-top scenario with various placements of objects. I understand that building a language grounding system on a real robot, and performing a user study is very hard, and I wouldn't want these obstacles to stand in the way of putting an excellent idea and the user study results out there. At the very least, the paper should make these limitations abundantly clear, so as to direct follow-up work (perhaps from groups with language grounding expertise) to improve upon these aspects of the system.

---

> ### Author Response · Authors · 2021-08-31
> **General Response [TkuX]**
>
> We’re incredibly grateful for your thorough review. Like you, we’re incredibly excited by the follow-up work that this paper opens up, and we hope we can assuage any remaining doubts with this response, and the top-level comments above.
>
> The biggest concern seems to be the following:
>
> > *“Unless I am missing something, the system does not take any visual inputs or observations of the environment as input. It is therefore incapable of actually understanding or grounding the natural language utterances to percepts or entities in the world... there does not appear to be a way to scale the system to a practical scenario, even a simple table-top scenario with various placements of objects”*
>
> We want to be transparent, and we know that there are many definitions of what constitutes “grounding” in our field. Like you mention, we *do not ground in visual or perceptual inputs* — any grounding we do, if you can call it that, is through the joint states of the robot. This is some instantiation of grounding (the robot learns to correlate some sequences of tokens with actual joint states — e.g. “fruit basket” to joint state [0.2, 0.23, 0.9, 0.43, 0.54, 0.11, 0.89]), but in terms of capabilities, you’re completely correct. If the objects moved at test time, or a user tried to provide an instruction not already covered, we’d fail. We make these limitations clear in the paper, following your suggestions.
>
> Like you say, building a language grounding system on a real robot is hard and has challenges (see the top-level comment around the Imitation Learning baseline, and associated webpage for more of these challenges), but *we really hope that you can see the merit in this work — the potential of language for multi-task disambiguation — despite these limitations*. Incorporating language for just disambiguation in the latent actions paradigm was an unknown from a research perspective; we weren’t sure if we could use language to learn multiple task-specific control spaces, how much data it would require, and so on. Now that we’ve shown the possibilities just with disambiguation, we hope that future work will go the way of different types of linguistic and environment generalization, like the prior instruction-following methods you mention (it’s not clear what work exactly you are thinking of here, but the work we know that has these features often require hundreds to thousands to hundreds of thousands of demonstrations or episodes for reinforcement learning — e.g., [Anderson et. al. 2018](https://arxiv.org/abs/1711.07280), [Blukis et. al. 2019](https://arxiv.org/abs/1910.09664), [Stepputtis et. al. 2020](https://arxiv.org/abs/2010.12083).
>
> Concretely, to address the last question about scaling to a practical scenario, we have already been taking cues for incorporating vision from the [Visual Latent Actions](https://arxiv.org/abs/2105.00580) work  for full perception (using CLIP like you suggest, over YOLO-v5). In the meantime, we have started playing with AR Tags + simple RGBD cameras to do object position tracking, which doesn’t hurt sample efficiency much in our preliminary experiments — both these threads indicate that scaling is possible (but for this work, beyond the scope).
>
> ---
>
> We also hope we can respond to the other concerns:
>
> > *“Annotator typicality & best annotators.”*
>
> We hope this is made more clear from the top-level comment around annotations and filtering — we’re just removing annotators producing SPAM.
>
> > *“Mean-pooled Glove vectors...”*
>
> We tried this first! We found that these were pretty decent, but the Distil-Roberta model was much better, and didn’t hurt us much from a compute perspective. This probably comes down to the contextual nature the model.
>
> > *“More model details...”*
>
> We did not get to this for the rebuttal-revision mainly to address the other big experimental and conceptual issues raised in the reviews, but we will make these changes for the camera-ready.
>
> > *“What guarantees that the controller exposes an intuitive action space that listens to the user?”*
>
> With LILA, the robot only acts when the user acts — not listening to the user isn’t a possibility in that case (e.g., there's no circumstance where a user leaves the controller idle, and the robot goes on to execute some motion/solve the task/do anything — the latent action model is only “active” when a user input has been registered). While decoder mode collapse is a possibility, we leverage tips and losses encompassed by the original latent actions work ([Losey et. al., 2019](https://arxiv.org/abs/1909.09674), [Li et. al., 2020](https://arxiv.org/abs/2007.11627)) to encourage intuitive control — specifically, we restrict the model from mapping z = 0 (no input from the user) to anything other than the null action, via an auxiliary loss penalty. As a result, only "meaningful" z inputs (inputs where the user is actually manipulating the controller) move the robot — users have full control and the model cannot ignore their inputs.

---

### Official Review · Reviewer_dhc1 · 2021-07-28

**Originality:** Good
**Technical Quality:** Fair
**Clarity Of Presentation:** Good
**Impact:** 3

**Recommendation:**

Weak Reject: I recommend rejecting the paper, but will not argue for my recommendation if the majority of other reviewers have a different opinion.

**Summary:**

The paper proposes a shared autonomy framework that enables people to control a robot arm using a combination of natural language text and traditional (joystick) control input. While most language understanding algorithms for robots operate in the full autonomy regime, the ability for people to also provide lower-level inputs helps to mitigate deficiencies in the autonomy (or the language understanding) and provides an important sense of agency. The proposed framework encodes state-control-language tuples into a learned latent action space that is then decoded to generate low-level actions for the robot. The model is trained on a dataset state-action pairs that were subsequently annotated. At test time, a given utterance is mapped to the nearest utterance in the training set using a pre-trained language model, which is then provided to the decoder. The method is evaluated on different manipulation tasks and compared with end-effector and imitation learning-based language understanding baselines. The proposed framework outperforms both baselines in terms of task success. A user study further assesses the qualitative benefits of the approach.

**Issues:**

In addition to the comments/questions above:

* Did the experiments involve a real robot arm? I assume that they did, but the text is a bit unclear.
---> Clarified in author response
* Are one-to-one state-action-utterance mappings needed (line 164) and if so, why is it reasonable to assume that there is an utterance for every state-action pair? If not, at what level do they need to be aligned?
---> Clarified in the author response
* Did the authors consider an IL baseline that only used control input (as opposed to only language)?
---> The author response largely answers my questions, but it should be clarified in the main text and not just the supplementary material.


**Reviewer Expertise:**

Excellent: Expert knowledge on the topic of the paper

**Strengths And Weaknesses:**

STRENGTHS:

* The idea of coupling natural language-based control with lower-level inputs in a shared autonomy framework is interesting.
* The paper presents the results of a user study that demonstrate the qualitative differences between the proposed method and standard baselines.


WEAKNESSES:

* The primary advantage of combining language with lower-level inputs is the ability to leverage their complementary nature. While there may be many tasks for which language would be a sufficient way for a user to convey their intent, there are others for which another means of input is necessary (particularly in partially observable/unstructured settings). However, the scenarios that are considered here are fully known and quite structured.
* The poor performance of the language-only baseline is suspicious. I imagine that it is a result of the fact that the state does not include information about the environment, in which case one can't expect a language-only version to perform the manipulation tasks well. Given the demonstrated success of language understanding for navigation and manipulation, a more standard baseline would presumably do much better.
* I like the idea of using language to influence learned latent actions as suggested by the title, however there is very little discussion and no evaluation of how the learned actions are influenced by language.
* The architecture combines existing components in an vanilla manner. The novelty is unclear.
---> The author response helps to clarify the novelty of the method.
* In addition to the work on latent actions, there is relevant work on shared autonomy in the case of unknown goals [1], including approaches that are task-agnostic [2].
---> The author response acknowledges this work and discusses the ways in which they differ
* The paper omits a large body of work in language understanding, including non-neural methods for language grounding in known and unknown environments [3]. These approaches have been shown to be effective for challenging manipulation commands despite being trained on relatively few samples. Of course. this comes at the cost of having to hand-engineer features and having a means of resolving abstract symbols to low-level actions (e.g., in the form of a policy), but that functionality exists, at least in the context of the scenarios considered here.
---> Again, the author response notes the intention to place the proposed method in the context of this body of work.
* The paper should discuss relevant work that combines natural language understanding with gestures [4--9], as complementary input modalities.
* The paper could be clearer on the amount of supervision required for the proposed approach.
---> The author response addresses this concern.
* It isn't clear how the method can be expected to generalize without reasoning over the environment state.
---> I don't feel that this point was adequately addressed. Instead, it was left for future work.
* While hierarchy is noted in the discussion section, a key reason that existing language understanding models can be trained in a relatively efficient manner and are able to generalize is that they exploit the hierarchical nature of language. This is largely ignored in the text.
* Language retrieval in these settings is a function not only of the utterances, but also the environment, however the proposed approach performs retrieval only using language.


COMMENTS/QUESTIONS:

* Lines 54--60: The deficiencies of existing language understanding methods depends on the approach. Those that are given are certainly true of methods that ground language to low-level actions, but less so with those that employ more abstract symbols. In these cases, the advantage offered by incorporating low-level control seems more related to being able to tolerate scenarios when these abstract symbols (i.e., policies) can not be readily mapped to low-level actions.
* Line 200: It isn't clear how the proposed framework would behave given an utterance that significantly differed from those encountered during training with regards to making mistakes. What is to stop the architecture from effectively ignoring the user's commands or does it provide a mechanism that allows them to take full control. If so, how does this mechanism differ from one that you might integrate into a language-only system?
* The means by which annotators were filtered based on their similarity to the pre-trained model seems a bit unfair.
---> I mis-spoke in my initial review meant to comment on the possibility that filtering the training corpus based on self-consistency would reduce diversity.
* Line 303: The paper states that the latent actions model produces intuitive control spaces that humans can quickly grasp. Was this experimentally verified?



REFERENCES


[1] S. Reddy, A. Dragan, and S. Levine. Shared autonomy via deep reinforcement learning. RSS 2018

[2] C. Schaff and M. Walter. Residual policy learning for shared autonomy. RSS 2020

[3] F. Duvallet, K. T., and A. Stentz. Imitation learning for natural language direction following through unknown environments. ICRA 2013.

[4] C. Matuszek, L. Bo, L. Zettlemoyer, and D. Fox. Learning from unscripted deictic gesture and language for human-robot interactions. AAAI 2014

[5] T. Kollar, J. Krishnamurthy, and G.P. Strimel. Toward Interactive Grounded Language Acqusition. RSS 2013.

[6] C. Kennington, S. Kousidis, and D. Schlangen. Interpreting situated dialogue utterances: an update model that uses speech, gaze, and gesture information. SIGDIAL 2013

[7] S. Mohan, A. Mininger, J. Kirk, and J.E. Laird. Learning grounded language through situated interactive instruction. AAAI Fall Symposium 2012.

[8] D. Whitney, M. Eldon, J. Oberlin, and S Tellex. Interpreting multimodal referring expressions in real time. ICRA 2016.

[9] N. Mavridis. A review of verbal and non-verbal human–robot interactive communication. Robotics and Autonomous Systems 2015.

**Summary Of Recommendation:**

The primary contribution of the paper is the integration of language with low-level control in a shared autonomy framework. This is certainly an interesting line of research, particularly in the context of scenarios for which language and low-level control alone are insufficient. However, the significance of this contribution is unclear, in part because of questions about the experimental evaluation and the novelty of the architecture.

UPDATES BASED ON AUTHOR RESPONSE:

I thank the authors for their detailed response my comments/questions. Several of these have been resolved, but I still have concerns regarding the suitability of the language baseline and the (clarity of the) primary contributions of the paper, which is the use of language as an additional input modality to resolve ambiguity in joystick input. I do find this to be an interesting line of work, but the fact that including language outperforms the baselines that were used here for this set of arguably simple tasks isn't terribly surprising and it's not clear that the results would generalize to more challenging domains.

---

> ### Author Response · Authors · 2021-08-31
> **General Response [dhc1] - Part 1 / 2**
>
> Thank you so much for your thorough, detailed, and constructive review! We hope that we can address many of your comments here and in the updated text, as well as provide some clarifications. We’ll split our response into two parts — this first comment will focus on the general comments/clarifications, while our next comment will focus on Related Work.
>
> We first hope to provide some **clarifications**:
>
> > *“Did the experiments involve a real robot arm? I assume that they did, but the text is a bit unclear.”*
>
> Yes, all experiments and user studies were performed on a real Franka Emika Panda Arm. We have made this more explicit in the text so it's more clear, and we hope the figures/videos show this as well.
>
> > *“The poor performance of the language-only baseline is suspicious. I imagine that it is because the state does not include information about the environment... a more standard baseline would presumably do much better.”*
>
> We hope that this is addressed in our top-level comment around the language-conditioned Imitation Learning baseline. Our language-only baseline *is indeed* conditioned on the joint state of the robot, the exact same state input that our LILA method is provided with. The poor performance is a separate question, that is answered similarly on the attached webpage above; the punchline is that due to our current setup and low-data regime, Imitation Learning is not *sample efficient* enough to succeed.
>
> This fits with several pieces of prior work, including those that you’ve linked — the shared autonomy baselines ([Reddy et. al., 2018](https://arxiv.org/abs/1802.01744), [Schaff and Walter, 2020](https://arxiv.org/abs/2004.05097)) pretrain policies via RL on hundreds to thousands of episodes, and many of the language-conditioned baselines use symbolic methods with hand-crafted features and lots of data (e.g., the reference suggested [Duvallet et. al. 2013](https://ieeexplore.ieee.org/document/6630702) [3], requires 16-30 “directions” trained via 25 iterations of DAgger — an effective 400+ demonstrations, relative to the < 100 in our work). We hope this explanation suffices, especially weighed against the extra engineering problems with the real robot mentioned on the supplementary webpage — language-conditioned imitation learning *is a standard baseline*, is state-conditioned, and semantically “tries” to do the right thing (succeeds at task grounding), but is not efficient enough to learn to fully execute the tasks.
>
> > *“...there is very little discussion and no evaluation of how the learned actions are influenced by language... The paper states that the latent actions model produces intuitive control spaces that humans can quickly grasp. Was this experimentally verified?*
>
> We hope this is a misunderstanding — our user study tries to evaluate this directly. With the LILA model, language helps to directly condition the learned latent actions that a user is provided to complete a task. Without language (and this is addressed by the “no-language” baseline top-level comment above), a user wouldn’t be able to get the “correct” latent action space to make progress on the task. The entirety of our evaluation and discussion hinges on the influence language has over learning separate latent actions for tasks.
>
> > *“Did the authors consider an IL baseline that only used control input (as opposed to only language)?”*
>
> This is a question many reviewers have asked, so we responded to this in the top-level comment above around the “No-Language Baseline.” We included two figures — Figure 3(a) and 3(b) — in our original supplemental that show the random, non-intuitive behavior we get from the no-language latent actions model; however, we know now that those figures themselves were not enough. We hope the above comment, paired with the [extra videos showing the non-intuitiveness of the baseline (and its inability to make progress and solve *any* tasks)](https://sites.google.com/view/lila-corl21/home/rebuttal-no-language-baseline?authuser=0#h.rd53jubxf8w7) coupled with the formal argument around the number of degrees-of-freedom required help justify the omission.
>
> Punchline: the no-language baseline trained on all demonstrations with no conditioning learned to “average” the demonstrations for the multiple tasks, thus learned a “clobbered”/”overloaded” latent space that prevented users from making *any* progress. It was a vacuous baseline, and we’re happy to show more examples in the final paper if desired.

---

> > ### Author Response · Authors · 2021-08-31
> > **General Response [dhc1] - Part 2 / 2**
> >
> > > *“The architecture combines existing components in a vanilla manner. The novelty is unclear.”*
> >
> > To our knowledge, while FiLM has been used in reinforcement learning and some robotics tasks, combining a FiLM-based autoencoder for combining language with robot control is novel, especially in the context of shared autonomy. Even if the architecture is simple, there is a lot of value to the task itself. That straightforward, easy-to-implement methods can work on this novel task is hopefully a clear benefit, not a drawback.
> >
> > > *“The paper could be clearer on the amount of supervision required for the proposed approach.”*
> >
> > Section 5 goes over the details for the amount of supervision required for LILA — Demonstration Collection denotes the amount of demonstrations collected per task for LILA (15 x 5 tasks). We additionally qualify how these demonstrations are collected (kinesthetically) and how many language annotations we collect in the sub-section “Crowdsourcing Language Annotation.” Please let us know if there are any details that you suggest we add in this section.
> >
> > > *“The means by which annotators were filtered based on their similarity to the pre-trained model seems a bit unfair.”*
> >
> > We hope this is another misunderstanding that we can clear up. The top-level comment regarding “Annotation & Filtering” tries to make this precise — we *do not* filter annotations based on similarity to a pre-trained model at all! Instead, we filter out SPAM utterances by looking at clusters of annotations for a given task, averaged *across users*. Some examples of the SPAM we catch can be found above — we’re not trying to filter out anything close to meaningful data here. We’re happy to provide additional experiments for the final camera-ready if you think that would be helpful.
> >
> > ---
> >
> > We now hope to answer remaining **questions:**
> >
> > > *“Are one-to-one state-action-utterance mappings needed (line 164) and if so, why is it reasonable to assume that there is an utterance for every state-action pair? If not, at what level do they need to be aligned?”*
> >
> > This work focuses solely on task-disambiguation using language (e.g., language specifies a single goal). This means that we only need to collect a language instruction *per demonstration* (not per state-action pair!). This is extremely sample efficient. However, we realize that scaling to multi-resolution language (“Pick up the coffee cup... no the one on the left... a little bit closer”) would require more data. We believe multi-resolution language is outside the scope of this paper, and we hope to address this in our future work.
> >
> > > *“What is to stop the architecture from effectively ignoring the user's commands...*
> >
> > With LILA, the robot can only act when the user acts — the idea of not listening to the user isn’t a possibility in that case (e.g., there's no circumstance where a user leaves the controller idle, and the robot goes on to execute some motion/solve the task/do anything — the latent action model is only “active” when a user input has been registered). At the base of the LILA model is a *conditional* auto-encoder. While decoder mode collapse is a possibility, we leverage tips and losses encompassed by the original latent actions work ([Losey et. al., 2019](https://arxiv.org/abs/1909.09674), [Li et. al., 2020](https://arxiv.org/abs/2007.11627)) to encourage intuitive control — specifically, we restrict the model from mapping z = 0 (no input from the user) to anything other than the null action, via an auxiliary loss penalty. As a result, only "meaningful" z inputs (inputs where the user is actually manipulating the controller) move the robot — users have full control and the model cannot ignore their inputs.
> >
> > > *“The primary advantage of combining language with lower-level inputs is the ability to leverage their complementary nature...*
> >
> > This work — incorporating language for just disambiguation in the latent actions paradigm was an unknown from a research perspective; we weren’t sure if we could use language to learn multiple task-specific control spaces, how much data it would require, and so on. Now that we’ve shown the possibilities just with disambiguation, we hope that future work will go the way of different types of linguistic and environment generalization, like the instruction-following methods you mention (beyond hand-engineered features, but exploring the same types of problems around compositionality, referring expressions, and the like), multi-resolution language, learning new tasks adaptively, and so on.
> >
> > We want to be transparent — we’re using language in a simple but incredibly useful form here, to disambiguate tasks. We agree that future work that realizes the full potential of language needs to move beyond that. But as an initial step, this work shows that language + latent actions is a viable strategy that is 1) sample-efficient relative to strong baselines, 2) intuitive for users, and 3) simple.

---

> > > ### Comment · Reviewer_dhc1 · 2021-09-02
> > > **Clarifications**
> > >
> > > Thank you again for the detailed response to my comments/questions:
> > >
> > > It seems that there may be a misunderstanding regarding my question about the system ignoring the user's commands. The comment/question ("It isn't clear how the proposed framework would behave given an utterance that significantly differed from those encountered during training with regards to making mistakes. What is to stop the architecture from effectively ignoring the user's commands or does it provide a mechanism that allows them to take full control.") wasn't asking about the robot taking action without any user input, but rather about what actions it would take when given user input that differed from that encountered in training (i.e., "*effectively* ignoring the user's commands).
> > >
> > > I appreciate the authors' clarification regarding filtering. I mis-spoke in my initial review and did not mean to say that they were filtered according to the pre-trained model, but rather that they seemed to have been filtered to ensure self-consistency, thereby limiting the diversity of the training corpus. I understand that this helps to remove outliers, but perhaps some of these outliers could actually be valid descriptions of the task (i.e., not noise). Then again, this would only make generalization harder, so the fact that it works at test time is encouraging.
> > >
> > > I'm curious the extent to which the model is able to lock onto keywords that allow it to identify the task (i.e., turning it into a five-way multiclass classification problem), where by receiving an instruction with the word "banana" would identify a Pick Banana task or the words "pour" and "cereal" suggest it is a Pour Bowl task.
> > >
> > >
> > > The original text states that: "... we filter data to identify workers who provided “atypical” annotations. We defined “atypical” by measuring the cosine distance between the sentence embedding (per the pre-trained language model from of an annotator’s provided sentence, and the average sentence embedding aggregated over all annotators for the given task."

---

> > > > ### Author Response · Authors · 2021-09-02
> > > > **Response to Clarifications**
> > > >
> > > > Apologies for the misunderstanding - let us try to rephrase:
> > > >
> > > > > *... wasn't asking about the robot taking action without any user input, but rather about what actions it would take when given user input that differed from that encountered in training*
> > > >
> > > > This is a good question; due to the inductive bias introduced by the *retrieval* process, we will always map a user utterance (even nonsense utterances) to the nearest neighbor in the training set – it'll be this utterances that are fed. So at test time the latent actions model *effectively only sees utterances its seen before*. This is definitely a limiting assumption, and there are ways to prevent the language model/robot from acting *at all* (e.g., say "I didn't understand") in light of completely novel utterances – see [Karamcheti et. al., 2020](https://arxiv.org/abs/2010.05190) for a threshold-based approach that does this – but for our setting this was an assumption that allowed for enough generalization to run a successful user study.
> > > >
> > > > > *I'm curious the extent to which the model is able to lock onto keywords that allow it to identify the task..."*
> > > >
> > > > This is a great question; our embeddings right now are sentence-contextual, but we're happy to a saliency-type analysis on word embeddings (e.g., mean-pooled GloVe embeddings) to actually see what the models are picking up on for the final paper!

---

> > > > > ### Comment · Reviewer_dhc1 · 2021-09-02
> > > > > **Role of keywords**
> > > > >
> > > > > Thanks for the prompt response.
> > > > >
> > > > > > This is a great question; our embeddings right now are sentence-contextual, but we're happy to a saliency-type analysis on word embeddings (e.g., mean-pooled GloVe embeddings) to actually see what the models are picking up on for the final paper!
> > > > >
> > > > > I wouldn't be surprised if this is what is going on with the joystick input helping to further resolve the task.

---

> > ### Comment · Reviewer_dhc1 · 2021-09-02
> > **Clarifications**
> >
> > Thank you for the clarifications.
> >
> > > We hope that this is addressed in our top-level comment around the language-conditioned Imitation Learning baseline. Our language-only baseline is indeed conditioned on the joint state of the robot, the exact same state input that our LILA method is provided with. The poor performance is a separate question, that is answered similarly on the attached webpage above; the punchline is that due to our current setup and low-data regime, Imitation Learning is not sample efficient enough to succeed.
> >
> > My point is that other than settings in which the environment doesn't change from training to test, it is seemingly impossible that a language-only baseline with access to only the joint states of the arm (i.e., no observations of the environment) would do well at these tasks. The only exception would be if the language provided low-level commands, effectively acting like a verbal joystick.
> >
> > > This fits with several pieces of prior work, including those that you’ve linked — the shared autonomy baselines (Reddy et. al., 2018, Schaff and Walter, 2020) pretrain policies via RL on hundreds to thousands of episodes, and many of the language-conditioned baselines use symbolic methods with hand-crafted features and lots of data (e.g., the reference suggested Duvallet et. al. 2013 [3], requires 16-30 “directions” trained via 25 iterations of DAgger — an effective 400+ demonstrations, relative to the < 100 in our work). We hope this explanation suffices, especially weighed against the extra engineering problems with the real robot mentioned on the supplementary webpage — language-conditioned imitation learning is a standard baseline, is state-conditioned, and semantically “tries” to do the right thing (succeeds at task grounding), but is not efficient enough to learn to fully execute the tasks.
> >
> > As noted in my response above, it was not my intention to suggest that these works are superior to this work, but rather to ask for a qualitative discussion of their differences, which this and the other response do nicely. These methods access to (an albeit structured) model of the world or observations of the world, in which case there is signal with which one could expect a language-only baseline to extract the necessary information to complete the case. Unless I am missing something, that isn't the case for the experiments considered here.
> >
> > I'd also note that the nature of the demonstations differs between these works. While some may require a greater number of demonstrations, the demonstrations themselves are easier to provide.
> >
> > On a related note, can the authors provide further clarification on the sentences that were used to fine-tune the language model? How many sentences were used? Where were they sourced?
> >
> > > “...there is very little discussion and no evaluation of how the learned actions are influenced by language... The paper states that the latent actions model produces intuitive control spaces that humans can quickly grasp. Was this experimentally verified?
> >
> > These are actually two distinct points.
> >
> > With regards to the first point ("I like the idea of using language to influence learned latent actions as suggested by the title, however there is very little discussion and no evaluation of how the learned actions are influenced by language"): It was more a comment of what the authors mean by "influenced"---i.e., can it be shown that the learned actions are semantically different from those learned without actions? I understand that the resulting behavior is different since the inputs are different, but this is to be expected.
> >
> > Regarding the second point ("The paper states that the latent actions model produces intuitive control spaces that humans can quickly grasp. Was this experimentally verified?"): The results do show that LILA offers several qualitative advantages, but in terms of "task ease" and "effortless", which seem most correlated with how quickly users grasped the interface, the differences aren't as significant.

---

> > > ### Author Response · Authors · 2021-09-02
> > > **Response to Clarifications**
> > >
> > > Thank you again for discussing this with us!
> > >
> > > > *My point is that other than settings in which the environment doesn't change from training to test, it is seemingly impossible that a language-only baseline with access to only the joint states of the arm... would do well at these tasks.*
> > >
> > > We completely agree, and there seems to be a misunderstanding here; we **only focus on the fixed environment setting here** because our goal was to explore the value of language for task disambiguation. If we were to change the environment state, we agree that a joint-state only representation *would not* be enough – for both imitation learning (language-only) **and** LILA (both would fail with joint state only, if the environment changes). But as you remark, as environments are identical across train and test, the comparison between LILA and Imitation Learning is apples-to-apples right now; the *failure of imitation learning is solely due to the sample efficiency argument above*.
> > >
> > > > *These methods access a model of the world... in which case there is signal for a language-only baseline to extract the necessary information to complete the [tasks] ... Unless I am missing something, that isn't the case for the experiments considered here.*
> > >
> > > Like the above, we agree that in dynamic settings where environments change from train --> test, you need extra perception/structure to ground objects. However, our experiments assume fixed environments -- and like you say in your top point, under this assumption, it's a fair comparison between language-only and LILA (since the environment *does not change*). We know this may not be entirely realistic, but from a research perspective, we hope you can see the value in starting with a targeted contribution, eliminating possible confounds.
> > >
> > > The truth is, we didn't know if adding language for disambiguation would work for latent actions. Rather than jump straight to perception, we wanted to explore solely this functionality with this paper, and hope we have succeeded in showing the potential for language in these settings. Future work (and already is) relaxing this fixed environment assumption by incorporating perception into state representations, which is already proving to add more complexities beyond the scope of *this* work.
> > >
> > > ---
> > >
> > > > *Can the authors provide further clarification on the sentences that were used to fine-tune the language model?*
> > >
> > > We *did not fine-tune* the language model ourselves. We used a default Distil-Roberta model fine-tuned on sentence paraphrases from the `sentence-transformers` library: [https://huggingface.co/sentence-transformers/paraphrase-xlm-r-multilingual-v1](https://huggingface.co/sentence-transformers/paraphrase-xlm-r-multilingual-v1). The data sources can be found [here](https://www.sbert.net/examples/training/multilingual/README.html#sources-for-training-data). We used this model out of the box. Both LILA and the language-only baseline use the same utterances/retrieval process based on this model, which can be found in our supplement.
> > >
> > > > *...can it be shown that the learned actions are semantically different from those learned without actions [we think you mean language here - let us know if not the case]?*
> > >
> > > Apologies for conflating the two points. Yes, we hope that the above top-level comment around the "No-Language" baseline shows this. Language is critical for *task disambiguation* without, we see complete collapse of the latent actions model, resulting in a [controller that is *unable to make progress on tasks!*](https://sites.google.com/view/lila-corl21/home/rebuttal-no-language-baseline?authuser=0#h.rd53jubxf8w7).
> > >
> > > > *intuitive control spaces that humans can quickly grasp...the results do show that LILA offers several qualitative advantages, but in terms of "task ease" and "effortless... the difference aren't as significant."*
> > >
> > > I think there's a bit more to this story than solely these qualitative metrics. First, that the "task ease" metric significantly favors LILA over all other approaches, is actually pretty important. Effortlessness is less clear, you're absolutely right – but when paired with the *quantitative success rate metrics,* that LILA models obtain higher success rates *across the board* with a minimum-jerk experience also should play into this story of the fact LILA exposes intuitive spaces humans can quickly grasp; if they couldn't grasp the control space, we'd expect lower success rates across the board.
> > >
> > > ---
> > >
> > > We hope that the value of this research – in examining how language can be used to disambiguate tasks in a latent actions framework – is clear, and hopefully merits more than an outright rejection! Independent of a need for extra perception or evaluation across changing environments (which can add several other confounds), we think this work lays a solid foundation for a series of future work, and we hope we can share it with the community, and we hope you agree.

---

> > > > ### Comment · Reviewer_dhc1 · 2021-09-02
> > > > **Fixed environments**
> > > >
> > > > Thank you for clarifying.
> > > >
> > > > I suspected that there wasn't any difference between training and test. I wouldn't refer to settings in which there is a difference between training and test as being "dynamic", but rather a standard form of generalization that would be typical of any real-world setting. I appreciate the extent to which the model is able to use language as a means of disambiguating intent in this setting, but it isn't realistic and as noted before, I don't see how the lessons learned from this setting would generalize.
> > > >
> > > > > We hope that the value of this research – in examining how language can be used to disambiguate tasks in a latent actions framework – is clear, and hopefully merits more than an outright rejection! Independent of a need for extra perception or evaluation across changing environments (which can add several other confounds), we think this work lays a solid foundation for a series of future work, and we hope we can share it with the community, and we hope you agree.
> > > >
> > > > Yes, that is fair and I have updated my overall score. However, I still believe that the contributions are limited by the inadequacy of the baselines and the narrow scope of the experimental setting (i.e., limited number of tasks, no environment variation between training and test).
> > > >
> > > > Thinking about it more, I think that the argument could be made that the joystick input is actually resolving ambiguities in language.

---

> ### Author Response · Authors · 2021-08-31
> **Related Work**
>
> We are grateful for these pointers to existing related work. Where applicable, we’ve added discussions both to the main text, as well as the supplemental (please let us know if you’d like us to rebalance this for the final camera-ready!). In the following, we hope to address and elaborate on some of your additional comments:
>
> > *“In addition to the work on latent actions, there is relevant work on shared autonomy in the case of unknown goals [1], including approaches that are task-agnostic [2].”*
>
> Both [Reddy et. al., 2018](https://arxiv.org/abs/1802.01744) [1] and [Schaff and Walter, 2020](https://arxiv.org/abs/2004.05097) [2] are great works in shared autonomy. We include these as helpful context in our discussion of related work. However, one thing we hope to point out and make clear so there is no confusion is that an apples-to-apples comparison of our work and this work is difficult; both these works, as well as prior work on policy blending assume pre-trained “base” policies that are trained to convergence (optimistically, requiring hundreds of episodes of experience for RL algorithms) prior to the shared autonomy loop, whereas our approach learns only from a handful of demonstrations.
>
> > *“The paper should discuss relevant work that combines natural language understanding with gestures [4--9], as complementary input modalities.”*
>
> This is great work that we have included in our updated revision.
>
> > *“The paper omits a large body of work in language understanding, including non-neural methods for language grounding in known and unknown environments [3]. These approaches have been shown to be effective for challenging manipulation commands despite being trained on relatively few samples. Of course this comes at the cost of having to hand-engineer features and having a means of resolving abstract symbols to low-level actions (e.g., in the form of a policy), but that functionality exists, at least in the context of the scenarios considered here.”*
>
> You’re completely right in that we omit this work, and we shouldn’t have — we believe this further helps contextualize our approach, so we have added this to our revision. But like you mention, comparison to this work comes at the cost of hand-engineering features, or other assumptions. For example, we also have added discussion of work that maps language to logical reward functions requiring dynamics models to facilitate planning ([Arumugam et. al., 2017](https://arxiv.org/abs/1704.06616)). We also discuss more recent work that learns [language-conditioned policies in a more end-to-end fashion](https://arxiv.org/abs/1910.09664) at the cost of thousands of demonstrations and/or RL episodes.
>
> Even the reference you suggest — [Duvallet et. al. 2013](https://ieeexplore.ieee.org/document/6630702) is a bit misleading in the amount of data they require for their approach. They use hand-engineered features, and require a training set of ~16 - 30 demonstrations, but they make a strong assumption they can run 25 iterations of DAgger, bringing the “effective” number of demonstrations in their training set up to ~400 (at a minimum) — on top of an already heuristic, hand-coded semantic parser. The action space is also much simpler than the real-time, continuous 7-DoF action space we use in LILA, requiring discrete traversal over seen nodes in a navigation environment.
>
> We want to emphasize that there are differences across all these different types of prior work, and that “not all data is created equal.” We hope our above argument + videos of the imitation learning ablation paint a picture of how hard it is to get even a simple imitation learning agent to scale in our hardware setup, on real robots, on moderate manipulation tasks, and show even more powerfully that LILA and shared autonomy is a great, robust method that opens doors for future work.

---

> > ### Comment · Reviewer_dhc1 · 2021-09-02
> > **Updated Related Work**
> >
> > I appreciate the authors' detailed response and the updates to the paper to include references to these papers.
> >
> > To be clear, I did not mean to suggest that the authors should experimentally compare to these methods or that previous work was in all ways superior to the proposed method. My point was instead that the paper should qualitatively differentiate their work, i.e., what are the relative advantages and disadvantages, in order to make the contributions clearer.
> >
> > Regarding Schaff et al., [2020],  Section V.A. implies that their base imitation policy is trained from a relatively small number of human demonstrations. No further demonstrations are necessary.

---

> > > ### Author Response · Authors · 2021-09-02
> > > **Further Clarification**
> > >
> > > Thank you so much for engaging with us, Reviewer `dhc1`. The clarification you make above is extremely helpful, and we hope that the discussion we added in the revision helps do this – of course, for the final paper we'll make sure to make these distinctions regarding advantages/disadvantages even more clear!
> > >
> > > For Schaff et. al [2020] – this is a hard comparison to talk about. Something we hope to make even clearer is the distinction between the assistive teleoperation based approaches like LILA, and blending approaches like Schaff et. al. For example, you're right in that the "surrogate human" policies are trained from some fixed human demonstrations – 9 humans x 100 episodes (900 demos) for Lunar Lander, and 14 humans x 30 episodes (420 demos) for the Drone Reacher task – this already feels like more of an investment than the <100 demos we needed for LILA.
> > >
> > > However, the *co-pilot* policies used in Schaff et. al. (Section V.A) for blending are trained against these surrogate policies via RL on a large number of episodes (whatever 100M timesteps computes to). It's really not clear how to compare these (or that they should be from a data perspective – if you can run a simulator for RL, then this is free!). We'll endeavor to tease apart these differences further in our revision.
> > >
> > > Thanks again for your willingness to discuss with us!

---

### Author Response · Authors · 2021-08-31
**Annotation & Filtering [All]**

Reviewers `dhc1`, `TkuX`, and `Eg7K` have clarification questions around our filtering process when collecting language instructions from crowdworkers:
- `dhc1`: *"The means by which annotators were filtered... seems a bit unfair."*
- `TkuX`: *“It is not stated whether the best annotators are typical or atypical..."*
- `Eg7K`: *"It could be argued, then, that the data has been curated for the algorithm... [c]an the authors clarify this methodology?"*

---

We believe there is a misunderstanding here, stemming from the clarity of this section of our paper. We have updated the main body in Section 5 — “Crowdsourcing Language Annotation” to clarify this point, with further details in the Supplemental.

We do *not* filter annotators based on similarity to the pre-trained model. Instead, we cluster *annotators* (using word embeddings as a proxy) and remove the annotators that *clearly deviate from the average annotator*.

This is solely a systematic way to filter out SPAM from crowdsourced annotations, since this is a common problem. Using the deviation to find "atypical" annotators (where atypical in this case means spammers — annotators producing nonsensical responses) helps us crack down on SPAM without introducing our own biases.

For further context, here are 3 randomly sampled annotations we filtered out (recall that a usual utterance is an instruction describing robot motion like "pick up the green bowl and place on the gray tray"):
- *"robots save workers from performing dangerous tasks. they can work in hazardous conditions, such as poor lighting, toxic chemicals, or tight spaces."*
- *"humanoid robots are robots that look like and/or mimic human behavior."*
- *"these robots usually perform human-like activities (like running, jumping and carrying objects), and are sometimes designed to look like us, even having human faces and expressions."*

As you can see, these are clearly SPAM annotations. Other types of SPAM that we crack down on with this procedure include other types of poor quality annotations, including just copy-pasting their first “valid” instruction across other videos, where the given instruction doesn’t make sense (though looks more valid). We hope this clarification assuages the current doubts, but we have additionally uploaded a new version of the supplemental material with a text file encompassing the full set of text we filtered out.

---

### Author Response · Authors · 2021-08-31
**No-Language Latent Actions Baseline [All]**

A question from meta-reviewer `Pima` and reviewer `o584` concerns the lack of a comparison to a baseline strategy that does not use language in our user study. Specifically:
- `Pima`: *"a central claim of the paper is that the addition of language is necessary to disambiguate the robot's actions... [h]owever, the evaluation does not include a comparison to an approach that does not use language."*
- `o584`: *"the results (though otherwise informative and convincing) lack a comparison to a learned method that does not use language. Such a comparison is important..."*

---

We first wish to clarify that we did include qualitative results from a no-language baseline in the supplemental material (Supplemental Figures 3a and 3b). However, we realize that this could easily be missed, and its omission from the user study was not immediately obvious. **We have assembled a more detailed response with qualitative videos of the no-language baseline & a formal argument at the following page** - *we hope the reviewers can take a look*: [https://sites.google.com/view/lila-corl21/home/rebuttal-no-language-baseline](https://sites.google.com/view/lila-corl21/home/rebuttal-no-language-baseline).

**To summarize the punchlines:**

The [qualitative video results](https://sites.google.com/view/lila-corl21/home/rebuttal-no-language-baseline?authuser=0#h.rd53jubxf8w7) ([see here for another example](https://sites.google.com/view/lila-corl21/home/rebuttal-no-language-baseline?authuser=0#h.je0q1cv5tne7)) show that it's next-to-impossible to get semantically meaningful behavior from a latent actions model that's trained on multiple tasks, but is only conditioned on state (no-language). *There is no way to disambiguate the task the user wishes to accomplish without language*, and as such, all the training demonstrations are "averaged," clobbering the latent action space — leading to unintuitive, random behavior that cannot make progress.

The [formal argument ](https://sites.google.com/view/lila-corl21/home/rebuttal-no-language-baseline?authuser=0#h.1iftjmvlrs8e) examines the "X" example from Figure 1 in our paper and goes on to show that beyond the fact that this model empirically acts randomly, it actually is inherently limited by the total "effective" degrees-of-freedom; it's just not possible to express all the behaviors we would want for humans to complete tasks with solely a 2-DoF action space.

In other words — this baseline would be *entirely vacuous had we included it in the user study*; it's impossible for users to make progress and solve tasks (both empirically and formally shown), so we omitted it. We hope this explanation suffices, and are happy to provide more examples if desired!

---

### Author Response · Authors · 2021-08-31
**Language-Only Baseline (Imitation Learning) [All]**

A critical concern from meta-reviewer `Pima` and reviewers `dhc1` and `Eg7K` is around the language-grounded/language-only baseline. Specifically there are a few comments we hope to address:
- `Pima`: *"The evaluation method does not appear to actually use language grounding as a baseline. The language only variant... does not actually perform any grounding"*
- `dhc1`: *"The poor performance of the language-only baseline is suspicious. I imagine that it is a result of the fact that the state does not include information about the environment..."*
- `Eg7K`: *"I was surprised to see End Effector performing better than Imitation Learning in some cases. Is there a clear intuition for this result? Maybe IL just needs a lot more demonstrations before the policy isn't too frustrating?"*

---

To clarify all of this, **we have assembled a detailed response & additional experiments at the following page** - *we hope the reviewers can take the time to read through our experiments*: [https://sites.google.com/view/lila-corl21/home/rebuttal-imitation-learning-baseline](https://sites.google.com/view/lila-corl21/home/rebuttal-imitation-learning-baseline).

**To summarize the punchlines:**

Our evaluation method *does use language-grounding as a baseline*. Our imitation learning agent is language-conditioned and state-conditioned, and behaves in a way that reflects an understanding of language.  It's conditioned on the same state of the environment as LILA and *learns to ground object demonstrations through joint space*, using the training demonstrations in a similar fashion as LILA. For example these videos for the tasks [“put the cereal bowl on the tray,”](https://sites.google.com/view/lila-corl21/home/rebuttal-imitation-learning-baseline?authuser=0#h.37m8ofh0k98y) and [“pour the blue cup into the coffee mug”](https://sites.google.com/view/lila-corl21/home/rebuttal-imitation-learning-baseline?authuser=0#h.l0ikj4ljsu3e) show that with 20 demos per task, imitation learning is clearly semantically grounding the right objective, but is unable to execute the fine-grained control necessary to obtain success.

As Reviewer `Eg7K` suggests, the intuition for why IL performs poorly is due to its *sample inefficiency*. This inefficiency emerges both from inherent noise when working with robots in general (imperfect resets, traditional cascading errors), *but also* from specifics in our real-world setup, where publishing actions (from a parent Python process) and reading states (from a child C++ robot controller) happen over a noisy channel, with a *drifting frequency* between publish/subscribe operations ([see our technical argument for more](https://sites.google.com/view/lila-corl21/home/rebuttal-imitation-learning-baseline?authuser=0#h.lpnx9k72mdyf)). We also present additional experiments thoroughly ablating imitation learning. We train with various amounts of data augmentation (3-5x more than LILA) and demonstrations (20/30 demos vs. LILA’s 10 per task — a 2-3x increase!) and show that while IL is able to semantically identify tasks and attempt to perform behaviors, it just cannot do so with the number of samples provided. For example, in this video of [“put the cereal bowl on the tray”](https://sites.google.com/view/lila-corl21/home/rebuttal-imitation-learning-baseline?authuser=0#h.37m8ofh0k98y) the robot is clearly grounding the correct objective — it’s reaching for the green cereal bowl, and then moving down towards the tray, but is unable to get close enough to the bowl to execute a grasp, due to cascading errors.

To provide further context relative to prior work (as mentioned by some reviewers), out-of-the-box comparison to alternative strategies is difficult; shared autonomy approaches like [Reddy et. al., 2018](https://arxiv.org/abs/1802.01744), [Schaff and Walter, 2020](https://arxiv.org/abs/2004.05097) pretrain policies for hundreds to thousands of episodes prior to performing updates, and additional work in language-conditioned imitation learning [Stepputis et. al. 2020](https://arxiv.org/abs/2010.12083), [Shridhar et. al. 2020](https://arxiv.org/abs/1912.01734) requires thousands of demonstrations *in simulation*. In contrast, our approach learns from 10s of real-world demonstrations (at a maximum).

---

### Author Response · Authors · 2021-08-31
**Summary [All]**

We're grateful to all our reviewers for their detailed feedback and questions, as well as to our meta-reviewer for summarizing the critical questions to address with this rebuttal. We appreciate the reviewers’ support for this idea – pairing shared autonomy with language – and we hope that our additional experiments (we hoped to be thorough with these, and it took a bit longer than we had thought — apologies!), clarifications, and technical arguments addressed the open questions and concerns. We're excited that the reviewers believe that this work *"explores a novel, relevant, and interesting idea... [o]pens doors to exciting follow-up work,"* and that they see benefit in our *"real-world end-user study, which is both rare and necessary to assess performance of systems with natural language interfaces."* Finally, we are really excited that the reviewers believe that when they *“imagine the intersection of "RoboNLP" and "CoRL", this paper is a great example."*

We have addressed each reviewer and the meta-reviewer's comments via individual "official replies," attempting to provide clarifications and additional evidence. Below, as top-level official comments, we've also provided top-level rebuttals to three arguments that showed up across reviewers: 1) the "lack" of a language-grounded (language-only) baseline, 2) the missing comparison to a no-language baseline, and 3) clarification about the language annotation procedure.

We're committed to making this paper the best that it can be, and we know you all are as well. These are great, insightful reviews, and we feel proud and grateful that you find this idea compelling; if there is anything else we can do, please let us know.

---

### Meta-Review · Area_Chair_Pima · 2021-08-15

**Recommendation:** Accept (Poster)
**Confidence:** 4

**Metareview:**

This paper introduces a shared autonomy input for manipulation that is augmented with language understanding such that discrete language inputs (classification of predetermined phrases) can be used to improve task performance.

Strengths:

Development of multimodal inputs for manipulation is an important research area.  The paper introduces a novel approach and includes a user study validation.


Weaknesses:

The evaluation method does not appear to actually use language grounding as a baseline. The language only variant being tested has no access to actual state data, and thus does not actually perform any grounding.  This significantly undermines the reported results.

Furthermore, a central claim of the paper is that the addition of language is necessary to disambiguate the robot's actions when trying to tackle a task in the presented shared autonomy scenario.  However, the evaluation does not include a comparison to an approach that does not use language.  This is an important result to include in order to demonstrate the added benefit of this additional modality.

The work is not well situated in existing literature on language understanding, please see notes by R1 for additional works to include in the discussion.

It is not clear how well the presented technique would generalize beyond the limited explored scenario.

The attached videos did not significantly add to the clarity of the paper.  Some annotation of their content would be helpful.

Please see detailed reviewer comments for a list of additional questions and suggestions.


Post rebuttal comment:  Thank you for your detailed responses to the reviewers.  The paper represents an interesting and valuable advance in the area of situated NLP, and the authors' additions and clarifications were very valuable.  As is clear from the comments, there were a few common areas of confusion/concern that require clear clarification in the paper:
* The term "language grounding" means different things in different research contexts, which initially caused confusion for majority of reviewers with regard to how this paper was written.  The authors should carefully define this term.
* Clear discussion of limitation with respect to generalization.  The state of the art for most ML techniques is that we test generalization in new scenes/applications/domains, and so again readers are coming in with this assumption in mind.  This should be clearly addressed in the paper, as well as how this current approach, which does not generalize to unseen objects, is a helpful research stepping stone.
* The baseline discussion needs to be clearly presented in the paper.

---

> ### Author Response · Authors · 2021-08-31
> **General Response [Pima]**
>
> Thank you for taking the time to distill all the reviewer's comments down to the core strengths and weaknesses. We hope that we have addressed all of the reviewer's individual concerns with the individual responses below, as well as the above general responses.
>
> We'd like to provide direct responses to the points raised in the meta-review here:
>
> > *“The evaluation method does not appear to actually use language grounding as a baseline."*
>
> We hope this is covered by the official comment regarding the imitation learning baseline above. We do have a language grounding baseline, and *it is standard*; it has full access to the state data (just like LILA), but as the above argument and associated webpage with experiments summarize, it is extremely sample inefficient.
>
> > *"However, the evaluation does not include a comparison to an approach that does not use language."*
>
> As mentioned in the other official comment above regarding the no-language baseline, *we do have qualitative examples of this baseline in our supplemental*, though we acknowledge it may not be enough to make this point.
>
> As such, we include [additional videos showing the no-language baseline is vacuous](https://sites.google.com/view/lila-corl21/home/rebuttal-no-language-baseline?authuser=0#h.rd53jubxf8w7), as well as a formal argument to the same. Essentially, the no-language baseline *does not allow a user to make progress on any task, as it has no mechanism to disambiguate intent*. This was obvious to us after implementing and testing it, which is why we didn’t explicitly discuss this but we realize this is something that should be in the main paper. We have updated the text to reflect this discussion in Section 5 and the Supplemental.
>
> > *"The work is not well situated in existing literature on language understanding."*
>
> Please see the individual comment under Reviewer `dhc1` labeled "Related Work." We agree there needs to be a broader discussion of language-conditional methods and have added it to our revision, but we also note key differences to prior work that make it hard for direct comparison.
>
> > *"It is not clear how well the technique would generalize beyond the limited explored scenario."*
>
> This paper shows that language is an effective tool for disambiguation, building a significant advance on top of prior work in shared autonomy (that either assumes fixed goal spaces, or limiting ways of specifying intent). Future work will combine this technique with other advances in perception and dynamic object tracking to build broadly capable systems for shared autonomy across dynamic environments that incorporate other facets of language (referring expression grounding, hierarchy, etc.). This includes other work in this space [that incorporates visual observations while remaining sample efficient](https://arxiv.org/abs/2105.00580) and principles from other work on [language-conditioned imitation learning](https://arxiv.org/abs/2010.12083).
>
> > *"The attached videos did not significantly add to the clarity of the paper. Some annotations of their content would be helpful."*
>
> This is great feedback; we agree that just having the videos in the supplemental without annotation isn't too useful. However, we hope that the accompanying website — [https://sites.google.com/view/lila-corl21/home](https://sites.google.com/view/lila-corl21/home) — is able to place the [videos in context](https://sites.google.com/view/lila-corl21/home?authuser=0#h.bi9zz98pspjz). Beside each video we annotate the actual instruction provided by the user (or the intended goal if End-Effector control).  We’ve removed the videos from the attached supplemental to prevent confusion.

---

> > ### Comment · Reviewer_dhc1 · 2021-09-02
> > **Sufficiency of the language-grounding baseline**
> >
> > > We hope this is covered by the official comment regarding the imitation learning baseline above. We do have a language grounding baseline, and it is standard; it has full access to the state data (just like LILA), but as the above argument and associated webpage with experiments summarize, it is extremely sample inefficient.
> >
> > I disagree with this point. Providing the language-only baseline with access only to the robot's state is not a fair comparison. As noted in my follow-up to the authors, with the exception of settings where the environment is identical between training and test or language is sufficiently low-level to provide joystick-like input, one can't expect the language by itself to be sufficient to carry out the task. A standard language baseline would have access to an observation of the scene (e.g., in the form of an embedding from a pre-trained network).
> >
> > > This paper shows that language is an effective tool for disambiguation, building a significant advance on top of prior work in shared autonomy (that either assumes fixed goal spaces, or limiting ways of specifying intent).
> >
> > Isn't the goal space fixed in this case? There are only five tasks that are known a priori, turning this into a five-way multiclass classification problem. I would expect an additional input modality (in this case language) to help with this classification problem. That is not to say that the idea isn't interesting, but rather that the claimed contributions seem to be overstated.

---

> > > ### Author Response · Authors · 2021-09-02
> > > **Clarification Regarding Language-Grounding Baseline**
> > >
> > > > *Providing the language-only baseline with access only to the robot's state is not a fair comparison. As noted in my follow-up to the authors, with the exception of settings where the environment is identical between training and test or language is sufficiently low-level to provide joystick-like input, one can't expect the language by itself to be sufficient to carry out the task.*
> > >
> > > We believe there's a serious misunderstanding here that we'd love to clarify again. We think it'll help resolve a lot of the comments!
> > >
> > > **In all our experiments, the train and test environments are identical.** As you note, language by itself *can be sufficient* to carry out the task solely from joint states under this setting. *Both* LILA and Imitation Learning only have access to joint states in our experiments - and if we were to change up the environments (new object locations across train/test), *both* approaches would fail.
> > >
> > > We know this by itself is a limiting assumption, but again, the truth is, we did not know that adding language just for disambiguation would work for latent actions. Rather than jump straight to perception, we wanted to explore solely this functionality with this paper, and hope we have succeeded in showing the potential for language in these settings.
> > >
> > > > *Isn't the goal space fixed in this case? There are only five tasks that are known a priori, turning this into a five-way multiclass classification problem.*
> > >
> > > The goal space is fixed, but the ways a human could *specify these tasks* are not; our user study is based on the principle that a user sees a video of a robot performing some task (no other language tags, or additional information!) and needs to perform the same task *using language and the 2-DoF* controller only.
> > >
> > > I think you're right in that this above response is maybe a little bit too much - I think more accurately (and we will update the paper to make this clear), we hoped to contrast this work with traditional work on [policy blending and follow-ups in shared autonomy](https://personalrobotics.cs.washington.edu/publications/dragan2012shared.pdf) that explicitly assume a fixed goal space, and use the user's actions to *update or parameterize belief over these categorical goals*. Language is a natural and expressive tool that explicitly replaces this sort of iterative belief modeling, letting users directly specify their intent. While fixed in this limited research setting (for scope), we hope this initial work opens the door to different types of generalization down the line.
> > >
> > > ---
> > >
> > > We're lucky to have a reviewer like you critically engage with us, and we hope that this discussion shows that we've thought critically about this work. We understand your concerns, truly -- but at the same time, we hope you can understand *why* we chose to scope this work the way we did, to avoid confounds, run clean user studies, and get clarity around the role of language for disambiguation on top of the latent actions framework. There's so much more to do, and we hope to explore these extensions (and hope that the community picks up on this as well!). But in terms of providing a foundation that shows incorporating language within shared autonomy frameworks like this, we hope you see the value – or enough to not reject it outright!

---

### Decision · Program_Chairs · 2021-09-13

**Decision:**

Accept (Poster)

**Comment:**

This paper introduces a shared autonomy input for manipulation that is augmented with language understanding such that discrete language inputs (classification of predetermined phrases) can be used to improve task performance.

Strengths:

Development of multimodal inputs for manipulation is an important research area.  The paper introduces a novel approach and includes a user study validation.


Weaknesses:

The evaluation method does not appear to actually use language grounding as a baseline. The language only variant being tested has no access to actual state data, and thus does not actually perform any grounding.  This significantly undermines the reported results.

Furthermore, a central claim of the paper is that the addition of language is necessary to disambiguate the robot's actions when trying to tackle a task in the presented shared autonomy scenario.  However, the evaluation does not include a comparison to an approach that does not use language.  This is an important result to include in order to demonstrate the added benefit of this additional modality.

The work is not well situated in existing literature on language understanding, please see notes by R1 for additional works to include in the discussion.

It is not clear how well the presented technique would generalize beyond the limited explored scenario.

The attached videos did not significantly add to the clarity of the paper.  Some annotation of their content would be helpful.

Please see detailed reviewer comments for a list of additional questions and suggestions.


Post rebuttal comment:  Thank you for your detailed responses to the reviewers.  The paper represents an interesting and valuable advance in the area of situated NLP, and the authors' additions and clarifications were very valuable.  As is clear from the comments, there were a few common areas of confusion/concern that require clear clarification in the paper:
* The term "language grounding" means different things in different research contexts, which initially caused confusion for majority of reviewers with regard to how this paper was written.  The authors should carefully define this term.
* Clear discussion of limitation with respect to generalization.  The state of the art for most ML techniques is that we test generalization in new scenes/applications/domains, and so again readers are coming in with this assumption in mind.  This should be clearly addressed in the paper, as well as how this current approach, which does not generalize to unseen objects, is a helpful research stepping stone.
* The baseline discussion needs to be clearly presented in the paper.